# When Responsibility Guidance Hurts: A Pilot Study of Pre-Execution Projection in LLM Agents

## Abstract

Multi-agent LLM orchestration is increasingly framed as a routing problem, an aggregation problem, or a post-hoc failure problem. We study an intermediate object that none of these frames opens up: the responsibility structure an agent projects between receiving a natural-language delegation and executing against an artifact. We formalize *responsibility projection* as multi-label weight prediction over a closed dimension set, instantiate it on $J_{v1.1}$ — a 12-dimension taxonomy for paper-research delegation (seven category dimensions, five cross-cutting) — and use the closure to make projections from different model families directly comparable. The primary empirical contribution (P1) is that pre-execution responsibility projection is measurable and family-attributable: on a 50-example pilot under the v1.3 Anthropic-excluded cross panel (`gpt-5` / `gemini-2.5-pro` / `grok-4`) with within-model variance estimated from five `claude-sonnet-4-6` repetitions at $T = 0.5$, cross-family projection mismatch is approximately $6\times$ within-family stochastic variance (median $R(d) = 5.87$, 95% bootstrap CI $[4.47, 7.96]$, paired bootstrap CI on $d_C - d_W$ excludes zero, Wilcoxon $p < 10^{-15}$), and the main-run extension at $n = 310$ gives median $R(d) = 5.40$ with CI $[4.86, 5.89]$. The secondary contribution (P2) is a negative actionability result: under a 12-judge Anthropic-excluded panel and a three-condition execution split, projection-driven execution shows a directional *disadvantage* relative to direct execution on the headline weighted-R1 settlement loss (paired diff direct_naive − projection_driven $= -0.139$, 95% bootstrap CI $[-0.169, -0.109]$), with the cost concentrating on R1.4 (novelty mapping) and R1.7 (citation audit) — the two dimensions whose $s = 5$ anchor demands deep specialty engagement; we report this as a boundary condition, not a refutation of the projection layer. The methodological contribution (P3) is a closed Stage 1 human-anchor pilot on R1.7 that surfaces an *anchor specifiability ceiling*: form-embedded sharpened anchors are insufficient for reliable rater application without a separately-read protocol document, and the human-anchor scalability constraint is anchor specifiability and domain-expertise gating, not rater throughput. The scope of this paper is one delegation category (R1, paper-research) at pilot scale ($n = 50$) with the P1 measurability extension validated at $n = 310$; the four-condition Experiment 2 with task-aware-routing and CLAMBER-style baselines, longitudinal reputation evaluation on real LLM agents, and a main-run-scale $r^\star$ extension are reserved for follow-up work.

## 1 Introduction

LLM-based agentic systems have moved from single-shot answer generators to delegated collaborators that perform consequential work — drafting research papers, reviewing code, interpreting results, and improving proposals (Gu, 2026; Cemri et al., 2025). Multi-agent variants extend this to teams of LLMs that decompose tasks, parallelize execution, and integrate diverse competencies (Maryanskyy, 2026). Yet collaborator-like behaviour does not by itself produce reliable collaboration: the dominant orchestration paradigms — routing, aggregation, post-hoc failure analysis — focus on either selecting an agent before or selecting an output after, and rarely interrogate what an agent has *committed to* between those two points. Recent work shows that interpretation quality (Mehta, 2026), ambiguity in user requests (Zhang et al., 2024; Pu et al., 2026),

miscalibrated uncertainty (Steyvers et al., 2025a; Xuan et al., 2026), and behavioural variance under identical prompts (Cui & Alexander, 2026) all matter for whether the executed work meets the user's actual need. What unifies these observations is that something happens between the natural-language delegation and the agent's first execution step — and the existing pipeline rarely exposes it.

A central source of this opacity is that natural-language delegations are usually *operationally underspecified*. A request like "Improve this paper draft." does not specify whether the agent should perform conceptual reconstruction, evidence-claim alignment, novelty positioning, or sentence polish; nor does it specify in what proportion. Several internally consistent executions of the same delegation lead to outputs that the user would judge differently. We call the agent's pre-execution choice across these alternatives a *responsibility projection*: a structured commitment, before any artifact is touched, to a particular subset of operational responsibilities. When the projection differs from the user's later-preferred responsibility structure, we have a *projection mismatch*. Existing routing decisions select an agent before this commitment; existing aggregation steps select among outputs after; existing post-hoc taxonomies label failures that the commitment may have caused. None of these pipelines makes the commitment itself measurable.

This paper makes the pre-execution commitment a first-class object. We formalize the projection $\pi(d, u, t, a) \to r$ as a multi-label weight prediction over a closed responsibility space $J_{v1.1}$ of $|J| = 12$ dimensions, partitioned into seven category dimensions (R1.1–R1.7) for paper-research delegation and five cross-cutting dimensions (RX.1–RX.5) that apply to any delegated work. We use the closure to make projections from different model families directly comparable, and we use that comparability to define and measure projection mismatch as cosine and Jaccard distance between projections. We then propose a closed-loop protocol — responsibility-bearing delegation — that projects $r$, solicits responsibility-bearing bids from candidate agents, triggers clarification when projection uncertainty is high relative to clarification cost, selects an agent under a responsibility-adjusted utility, executes, settles per dimension, and updates per-dimension reputation. The protocol is described in Section 4 and stated as Algorithm 1.

We instantiate the taxonomy and evaluate the projection layer (with a pilot projection-injection treatment for execution) on a 50-example dataset, AMBIGUOUSDELEGATION-50-R1, constructed from 32 synthetic and 18 modified-real artifacts under the closed taxonomy. The empirical contributions are: (i) projection mismatch is a measurable, family-attributable quantity at $T = 0$, exceeding within-family stochastic variance at $T = 0.5$ by roughly $6\times$ on the pilot (median $R = 5.87$, 95% bootstrap CI $[4.47, 7.96]$, paired bootstrap CI on $d_C - d_W$ excludes zero, Wilcoxon $p < 10^{-15}$); the main-run extension at $n = 310$ confirms the same direction (median $R = 5.40$, CI $[4.86, 5.89]$); (ii) under a redesigned 12-judge Anthropic-excluded panel and a three-condition execution split, projection-driven execution shows a directional *disadvantage* relative to direct execution on the headline weighted-R1 settlement loss, with the cost concentrated on R1.4 (novelty mapping) and R1.7 (citation audit) — the two dimensions whose $s = 5$ anchor demands deep specialty engagement; (iii) a closed Stage 1 human-anchor pilot on R1.7 independently surfaces a methodological boundary condition: form-embedded sharpened anchors are insufficient for reliable rater application without a separately-read protocol document, and the human-anchor scalability constraint is anchor specifiability and domain-expertise gating, not rater throughput. The two findings converge on R1.7 directly (Stage 1 provides a direct boundary-condition signal on R1.7); R1.4 is inferential, since R1.4 was deferred from the Stage 1 closure.

**Contributions**

- **Problem formulation.** We formalize multi-agent LLM orchestration as incomplete delegation under natural-language ambiguity, and we identify projection mismatch as a pre-execution failure mode that is structurally distinct from routing failure, aggregation failure, miscalibration, and post-hoc misclassification (Section 3).

- **Closed responsibility taxonomy.** We instantiate a closed 12-dimension responsibility taxonomy $J_{v1.1}$ for paper-research delegation; the closure is what makes pre-execution responsibility projections from different model families directly comparable. We additionally specify a closed-loop protocol (project → bid → clarify → select → execute → settle → reputation) for completeness (Section 4); this paper evaluates only the projection layer of that protocol.

- **P1 — Measurability (primary).** On 50 controlled examples under the v1.3 Anthropic-excluded cross panel (`gpt-5` / `gemini-2.5-pro` / `grok-4`) with within-model variance estimated from five `claude-sonnet-4-6` repetitions at $T = 0.5$, three distinct model families produce projections that differ from each other roughly $6\times$ more than the within-model baseline ($R(d)$ median 5.87); all five guard hypotheses pass at $p < 10^{-15}$. The main-run extension at $n = 310$ sharpens the same direction (median $R(d) = 5.40$, CI $[4.86, 5.89]$). Projection mismatch is measurable and family-attributable (Sections 6.2, 6.3).

- **P2 — Boundary condition on actionability (negative result, reported transparently).** Under a 12-judge Anthropic-excluded panel and a three-condition execution split, projection-driven execution scores worse than direct execution on the headline weighted-R1 settlement loss; the cost concentrates on R1.4 and R1.7. We propose one consistent explanation (broad-attention dilution on dim-specific deep work) and discuss two alternative explanations (format-injection, hard-dim null) that the pilot cannot fully separate (Sections 6.4, 6.5).

- **P3 — Anchor specifiability ceiling (methodological).** A closed Stage 1 human-anchor pilot on R1.7 surfaces a methodological boundary condition that any project proposing crowdworker human anchor as the validity backstop for an LLM-only annotator panel must address: form-embedded sharpened anchors are insufficient for reliable rater application without a separately-read protocol document. The convergence with P2 is direct on R1.7 (where Stage 1 applies) and inferential on R1.4 (not in the Stage 1 closure) (Sections 6.7, 7.8).

**Scope of this paper.** We instantiate the protocol on a single delegation category (R1, paper-research) with five cross-cutting dimensions, and we evaluate at pilot scale ($n = 50$) with one author and two LLM annotators on the hidden-intent track; the P1 measurability extension is validated on the main-run sample ($n = 310$), while the $r^\star$-weighted P2 settlement-loss analysis remains pilot-bound (Section 6.4.2, RX paragraph) because $r^\star$ is derived from the closed pilot human-annotation track. The other delegation categories in our long-term plan (code review, result interpretation, proposal improvement, technical summarization), the full four-condition Experiment 2 with task-aware-routing and CLAMBER-style baselines, the longitudinal reputation experiment on real LLM agents, and a main-run-scale $r^\star$ extension are reserved for follow-up work; their omission is disclosed throughout. The pilot's planned human-annotator extension was attempted, partially executed, and yielded a methodological boundary-condition finding on anchor specifiability, reported in Sections 6.7 and 7.8. The contribution of the present paper is that responsibility projection becomes *measurable* at this scope, naive projection injection is not yet actionable in current form, and human-anchor scalability is bound by anchor specifiability and domain-expertise gating; main-run extensions are the natural next step.

The rest of the paper proceeds as follows. Section 2 positions the work against adjacent orchestration, ambiguity, calibration, evaluation, and annotation frames. Section 3 fixes the closed taxonomy and the projection-mismatch quantity. Section 4 states the protocol and Algorithm 1. Sections 5–6 report the pilot setup and findings. Section 7 discusses implications and Section 8 concludes.

## 2 Related Work: Pre-Execution Commitment versus Adjacent Frames

We organize related work into ten substantive clusters plus a methodological reflexivity cluster and a brief summary. Each cluster occupies a slightly different temporal or structural slice of the agent pipeline. We position our contribution by contrast: the pre-execution responsibility commitment that this paper foregrounds is not the same object as task type, ambiguous information need, missing input, model confidence, behavioural variance, post-hoc failure mode, output-aggregation choice, judge bias, decomposed evaluation rubric, or annotator disagreement, even though every prior cluster shares some commitment with our framing.

## 2.1 Task-aware delegation, routing, and cost-aware model selection

The closest neighbour is task-aware delegation, which derives task-conditioned capability and coordination-risk signals from preference data and uses them to drive routing, auditor selection, and rationale disclosure (Gu, 2026). Its loop — task typing $\rightarrow$ capability profile $\rightarrow$ coordination-risk cue $\rightarrow$ routing decision $\rightarrow$ accountability log — shares our commitment to making delegation visible and auditable, to surfacing reliability cues to users, and to logging accountability after the fact. A complementary line of routing-and-cost work designs systems that pick a model per query under cost and quality trade-offs (Ong et al., 2024; Hu et al., 2024; Asenjo et al., 2023; Ding et al., 2024; Chen et al., 2023). Routing benchmarks such as RouterBench (Hu et al., 2024) stress-test these systems across heterogeneous tasks, and meta-modeling approaches learn the routing policy from preference data (Ong et al., 2024) or from cost/performance prediction labels (Asenjo et al., 2023). The shared commitment is that typed delegation is the right granularity. The difference is the typed object: routing systems condition on a task cluster $c$ and ask "which agent is reliable on this task type?", while we condition on a responsibility vector $r$ over a closed dimension set and ask "what responsibility structure has the agent projected from this delegation, and should it be clarified before commitment?" Routing returns a pointer; projection returns a structured commitment that can be partial, refused, or explicitly claimed.

## 2.2 Ambiguity identification and clarification

A long line of work studies whether language models can recognize and clarify ambiguous user queries. ClariQ (Aliannejadi et al., 2020) studies clarification in open-domain information-seeking dialogue with a benchmark of clarifying-question turns; CLAMBER (Zhang et al., 2024) more recently organizes ambiguity along epistemic-misalignment, linguistic-ambiguity, and aleatoric-output axes and tests whether language models generate useful clarifying questions. Open-domain QA work treats ambiguity as a target signal: AmbigQA constructs disambiguation pairs for ambiguous open-domain questions (Min et al., 2020), and learning-to-ask methods rank candidate clarification questions by expected information value (Rao & Daumé III, 2018). Recent work also asks when to clarify and when to answer: selective-clarification methods refuse only on the questions where clarification reduces error, leaving the rest answered (Lee et al., 2023; Cole et al., 2023). The shared commitment with our protocol is that clarification is an admissible mid-pipeline action; the difference is the object of clarification. Ambiguity benchmarks treat the user's information need as the ambiguous object ("when did he land on the moon?" — unclear referent). We treat the agent's responsibility commitment as the ambiguous object ("improve this paper draft." — unclear whether grammar polish, conceptual reconstruction, novelty audit, or all of these are being delegated).

## 2.3 Underspecification in long-horizon tasks

Long-horizon underspecification work generates controllably underspecified tasks by removing information along Goal, Constraint, Input, and Context axes, then validates the resulting variants behaviourally as outcome-critical, divergent, or benign (Pu et al., 2026). It also operationalizes clarification efficiency as a gain-per-question metric, internalizing the intuition that clarification is costly. We adopt the cost-sensitivity insight directly: the protocol's clarification trigger (Equation (11)) is conditioned on expected loss reduction net of clarification cost, not on ambiguity flags alone. The difference is what is missing: long-horizon underspecification studies missing task information; we study ambiguous responsibility boundaries on a delegation that may already specify the input fully.

## 2.4 Uncertainty communication and tool-use calibration

How an agent communicates uncertainty is a separate measurement track. Calibration work elicits confidence scores from language models and asks whether expressed confidence matches accuracy: verbalized-confidence prompts (Tian et al., 2023; Lin et al., 2022), internal-token-probability methods (Kadavath et al., 2022), and empirical evaluations across elicitation strategies (Xiong et al., 2024). A separate strand asks whether *users* accurately estimate model knowledge (Steyvers et al., 2025b), and a recent metacognition study examines whether supervised fine-tuning improves uncertainty communication and whether such improvement transfers

across tasks (Steyvers et al., 2025a). Tool-use calibration work disaggregates calibration further by tool type, observing that evidence tools (web search, retrieval) can induce overconfidence by surfacing noisy material as authoritative, while verification tools (code interpreters) ground reasoning through deterministic feedback (Xuan et al., 2026). We treat uncertainty as one cross-cutting responsibility dimension (RX.1) rather than as the main object, and we treat tool outputs as evidence whose reliability $\kappa(e)$ depends on source type (Equation (14)). The difference is scope: calibration work asks whether confidence matches correctness on the answer; we ask whether the agent disclosed uncertainty about the responsibility structure it claimed to assume.

## 2.5 Behavioural variance and reproducibility

Empirical studies show that an identical prompt and an identical task can produce substantially different analytical conclusions across models, prompting strategies, and temperatures (Cui & Alexander, 2026). Closely related, work on consistency in software-engineering agents shows that consistency correlates with accuracy — but also that consistency can amplify wrong interpretations, where many failures are "consistent wrong interpretation" rather than random error (Mehta, 2026). Even nominally deterministic settings exhibit non-determinism in LLM outputs across re-runs (Atil et al., 2024). The shared commitment is that variance and interpretation matter as much as raw accuracy. The difference is when in the pipeline we measure: behavioural-variance studies measure outcome and trajectory variance *after* execution; we measure responsibility-projection divergence *before* execution begins. Section 6.3 shows that pre-execution variance, captured as cross-family projection mismatch, is family-attributable rather than stochastic on the pilot dataset.

## 2.6 Multi-agent collaboration frameworks and failure taxonomies

Multi-agent LLM frameworks specify how agents collaborate to perform consequential work. MetaGPT structures agents as software roles with a meta-programming layer (Hong et al., 2024); AutoGen builds extensible multi-agent conversations (Wu et al., 2024); CAMEL studies role-playing communicative agents for cooperative task solving (Li et al., 2023); ChatDev simulates a software-development team (Qian et al., 2024); Generative Agents simulates social behaviour (Park et al., 2023); and multi-agent debate improves factuality through reasoned disagreement (Du et al., 2024). Recent work introduces post-hoc failure taxonomies for these systems, identifying categories such as system-design issues, inter-agent misalignment, and task-verification failures (Cemri et al., 2025). These taxonomies are derived from execution traces and label what went wrong after the fact. Among the named modes, "disobey task specification," "disobey role specification," and "fail to ask for clarification" overlap directly with our pre-execution governance concerns. The difference is temporal: collaboration frameworks coordinate already-typed roles, and failure taxonomies diagnose completed traces, while we govern the responsibility commitment that may or may not produce the trace in the first place. The frameworks and taxonomies are complementary to our protocol: a multi-agent framework could absorb the projection step as a pre-conversation contract, and the post-hoc taxonomy labels can serve as settlement evidence.

## 2.7 Aggregation and the selection bottleneck

When diverse multi-agent teams produce candidate outputs, the value of diversity depends on the quality of the aggregator: with a strong selector, diverse generation pays off; with a weak selector, it does not (Maryanskyy, 2026). A scaling-laws perspective asks whether more LLM calls reliably improve compound inference systems; the answer is dim-dependent and saturating (Chen et al., 2024). This line asks "given candidates, which output should be selected?" We instead ask "before any output is generated, which responsibility should be projected and committed to?" These are dual questions on opposite sides of generation. We include a generate-then-select baseline in the experimental design (deferred to the main run; Section 6.6) so that the cost of generating wrong-scope outputs and selecting among them can be compared directly to the cost of governing scope before generation.

## 2.8 LLM-as-judge bias and panel design

Using LLMs as judges for open-ended outputs introduces bias modes that the panel design must control. Position bias, verbosity bias, and self-enhancement bias are documented baselines (Zheng et al., 2023; Wang et al., 2023a). LLM judges are inconsistent across re-runs and biased in dataset-specific ways (Stureborg et al., 2024). Self-preference / narcissistic-evaluator effects appear when judges identify and favor their own outputs (Panickssery et al., 2024; Koo et al., 2024; Wataoka et al., 2024). Length-controlled evaluators correct one specific bias (verbosity) but do not eliminate the rest (Dubois et al., 2024). Replacing single judges with juries of diverse models is one mitigation (Verga et al., 2024); specialized evaluator models (Prometheus 2 (Kim et al., 2024), G-Eval with GPT-4 prompts (Liu et al., 2023)) are another. Our 12-judge tier-stratified Anthropic-excluded panel (Section 6.4) directly absorbs these mitigations: family exclusion controls self-preference, tier-stratification controls capability-tier confound, and panel size targets reliability. The difference from this cluster is the object being judged: judge-bias work asks "does the judge correctly compare two outputs?" while we ask "does the executor's responsibility projection match a panel-derived ground truth before execution?"

## 2.9 Decomposed and rubric-based evaluation

Rather than scoring an output as a single quality number, decomposed evaluation maps the output to per-skill or per-requirement components. FLASK decomposes into a fine-grained skill set and scores each skill independently (Ye et al., 2024); InFoBench decomposes complex instructions into per-requirement satisfaction (Qin et al., 2024); G-Eval scores along chained explanation steps (Liu et al., 2023); Prometheus 2 trains a dedicated evaluator language model on rubric criteria (Kim et al., 2024). Our closed responsibility taxonomy $J_{v1.1}$ shares the per-component structure: weights $r_j$ are predicted per dimension, and settlement is computed per dimension. The difference is the decomposition target: FLASK / InFoBench / G-Eval decompose the *output* (skills demonstrated, requirements satisfied, reasoning steps); we decompose the *pre-execution commitment* (responsibility dimensions projected). Per-component aggregation is the shared paradigm; the per-component object differs.

## 2.10 Annotator disagreement, human-label variation, and crowdworker reliability

Ground-truth annotation tracks have known reliability limits. Plank's framework reframes human-label variation as signal rather than noise (Plank, 2022), calling for distributional ground truths over single labels. Krippendorff's $\alpha$ formalizes inter-annotator reliability as the canonical measure for interval-level coding data (Krippendorff, 2011; Hayes & Krippendorff, 2007), and our $\alpha$ reporting (Section 6.1.3) follows this convention. The crowdworker-reliability literature documents that even careful Mechanical-Turk evaluations of generated text can systematically differ from expert evaluations (Karpinska et al., 2021; Clark et al., 2021), and recent observational work shows that crowdworkers increasingly use LLMs themselves to produce annotations (Veselovsky et al., 2023), with related work finding systematic variation in human-respondent data quality across online sample-provider platforms (Gordon et al., 2025). Our Stage 1 R1.7 closure (Section 6.7) confirms the crowdworker-reliability concern empirically: the Prolific pass rate against quality screening was 33%, and the dominant rejection pattern was AI-assisted submissions or task-misunderstanding. The difference between this cluster's framing and ours is what the human-anchor scalability constraint is taken to be: this cluster treats it as a recruitment / training / aggregation problem, while our boundary-condition finding (Section 7.8) identifies anchor specifiability and domain-expertise gating as binding even after recruitment and aggregation are well-handled.

## 2.11 Reasoning, tool use, and self-refinement

A separate line of work increases output quality by changing the executor's reasoning style: chain-of-thought prompting (Wei et al., 2022), self-consistency over multiple reasoning paths (Wang et al., 2023b), action-reasoning interleaving (Yao et al., 2023), tool-augmented generation (Schick et al., 2023), and iterative self-feedback (Madaan et al., 2023). Software-engineering work surveys how these techniques are applied to coding agents (Hou et al., 2024). The shared commitment is that the executor's intermediate steps matter;

the difference is what is being intermediated. These techniques alter how the executor reasons *within* an already-committed task; we alter what the executor commits to *before* reasoning begins. The two layers compose: an executor that has projected its responsibility could subsequently reason via chain-of-thought or self-refine, and our protocol does not prescribe a particular reasoning style.

### 2.12 Benchmarking the benchmarks

A reflexive thread asks what counts as a credible NLP benchmark (Bowman & Dahl, 2021). The recommendation is to fix construct validity, sample size, and analysis pre-registration before declaring an effect. Our pilot adopts $r^*_{\text{median}}$ pre-registered active-set rules (Section 5), pre-registered hypothesis tests with paired bootstrap CIs and Krippendorff's $\alpha$ (Section 5.5), and explicit reporting of negative results (Section 6.4). This cluster is methodological and shapes how we report; it is not a competing object.

### 2.13 Summary of differentiation

Across these ten clusters, the recurring contrast is the temporal locus of the typed object. Task-aware delegation and routing fix the typed object as a task cluster; ambiguity benchmarks fix it as a query gap; long-horizon underspecification fixes it as missing input; calibration work fixes it as confidence; behavioural-variance work fixes it as outcome variance; MAS frameworks and failure taxonomies fix roles or post-hoc failure modes; aggregation work fixes the candidate output; LLM-as-judge work fixes the comparison protocol; decomposed evaluation fixes the per-component output rubric; and annotator-disagreement work fixes the human-label distribution. Pre-execution responsibility commitment is the typed object that sits between natural-language delegation and execution, and the rest of the paper develops the closed-taxonomy formulation, the protocol, and the pilot evidence for measuring and acting on it.

## 3 Problem Formulation

We formulate multi-agent LLM orchestration as an *incomplete delegation* problem: a user issues a natural-language delegation that does not fully specify the operational responsibilities the executing agent must assume, and the agent's pre-execution responsibility commitment determines downstream success or failure. Section 3.1 fixes notation. Section 3.2 introduces the closed responsibility space $J_{v1.1}$ that this paper instantiates and evaluates. Section 3.3 defines the projection $\pi$ from delegations to responsibility weight vectors, and Section 3.4 defines the projection-mismatch quantity $M(d, u, t)$ that the experiments measure. Section 3.5 connects projection mismatch across model families to the empirical pilot in Section 6.2.

### 3.1 Notation

Let $d \in \mathcal{D}$ denote a natural-language delegation, $u$ the delegator, $t$ the task category or context, and $a$ the artifact supplied with the delegation. Let $A = \{a_1, \ldots, a_n\}$ be the candidate agent set and $y_i$ the role assigned to agent $i$. We write $J$ for the closed responsibility space, $|J|$ for its size, and $J_X \subseteq J$ for the cross-cutting subset. A weight vector $r \in [0, 1]^{|J|}$ over $J$ is a *responsibility projection*; its $j$-th coordinate $r_j$ is the projected importance of dimension $j$. We write $r^*$ for the hidden, annotated, or later-preferred responsibility vector that serves as the operational ground truth in controlled evaluation.

### 3.2 Closed responsibility space $J_{v1.1}$

This paper instantiates the protocol on $J_{v1.1}$, a closed taxonomy of $|J| = 12$ dimensions partitioned as

$$J_{v1.1} = J_{R1} \cup J_X, \quad |J_{R1}| = 7, |J_X| = 5. \tag{1}$$

The category subset $J_{R1}$ enumerates seven dimensions of paper-research delegation:

- R1.1 *Conceptual reconstruction* — reframe the thesis, gap statement, or contribution claim above the sentence level.

- R1.2 *Logical consistency* — verify the argument chain (premises, intermediates, conclusions, and cross-section coherence).

- R1.3 *Evidence-claim alignment* — check whether claims are supported by the experiments, tables, and figures as reported.

- R1.4 *Novelty assessment* — estimate the delta between the contribution and the nearest prior work.

- R1.5 *Structural reorganization* — section-level ordering and narrative arc.

- R1.6 *Writing polish* — sentence- and paragraph-level prose quality.

- R1.7 *Citation and scholarship* — reference accuracy, coverage of the relevant literature, and attribution.

The cross-cutting subset $J_X$ is treated as always-active by construction, regardless of $r$, because the five dimensions apply to any delegated work:

- RX.1 Uncertainty disclosure

- RX.2 Overclaim avoidance

- RX.3 Scope adherence

- RX.4 Downstream-harm avoidance

- RX.5 Provenance and traceability

**Why a closed taxonomy.** Open-vocabulary responsibility projection is unstable across raters and across model families: agreement on free-form "responsibility labels" is poor and the resulting $r$ vectors are not directly comparable. Closing the taxonomy at $|J| = 12$ converts projection from open generation to multi-label weight prediction over a fixed index set, enabling cosine and Jaccard distances between projections from different families to be computed and aggregated. Sections 6.2 and 6.3 report these cross-family and within-model distances on the pilot dataset.

**Scope of the closure.** $J_{v1.1}$ is the operational scope of this paper, not a universal taxonomy claim. The five additional task categories listed in our long-term plan (R2 code review and bug-fix, R3 result interpretation, R4 proposal improvement, R5 technical summarization) and any additional cross-cutting dimensions are reserved for future cycles. Section 6.6 discusses the implications of this scoping for the paper's claims.

### 3.3  Responsibility projection $\pi$

A projection agent maps the delegation context to a weight vector:

$$r \;=\; \pi(d, u, t, a), \qquad r \in [0, 1]^{|J|}. \tag{2}$$

Each weight $r_j$ takes values calibrated by four anchors — 0.0 "not requested," 0.3 "peripheral," 0.7 "central expected responsibility," 1.0 "load-bearing." Continuous values between anchors are permitted; the anchors fix the calibration. From $r$ the protocol derives the active set as the union of supra-threshold R1 dimensions and the always-active cross-cutting block,

$$J^*(d) \;=\; \{\, j \in J_{R1} : r_j > \tau_r \,\} \;\cup\; J_X, \tag{3}$$

with $\tau_r = 0.3$. The active set is the input to subsequent steps of the protocol (Section 4); its precision directly affects whether the executor concentrates on the true load-bearing dimensions or spreads attention across spuriously activated ones. Section 6.5.1 reports a prior hypothesis that a broader-than-engineered active set degrades execution quality, tested in the redesigned panel of this pilot; the active-set breadth excess turns out to be approximately equal across all three conditions and does not differentially explain the projection-injection effect. Section 4.9 discusses mitigations.

### 3.4 Projection mismatch

For controlled evaluation we suppose access to a hidden responsibility vector $r^*$ that may be operationally annotated, derived from clarification, or constructed as a later-preferred ground truth. *Projection mismatch* between an agent's projection $r$ and $r^*$ is

$$M(d, u, t) = \Delta(r, r^*), \tag{4}$$

with $\Delta$ a composite distance that penalizes the components a reviewer cares about. We use a four-component decomposition,

$$\Delta(r, r^*) = \alpha_{\cos}\big(1 - \cos(r, r^*)\big) + \alpha_{\mathrm{jac}} \operatorname{Jac}\big(J^*(r), J^*(r^*)\big)$$
$$+ \alpha_u \operatorname{UnderProj} + \alpha_o \operatorname{OverProj}, \tag{5}$$

with under- and over-projection defined on R1 dimensions only (RX is always active by construction):

$$\operatorname{UnderProj}(r, r^*) = \frac{|\{j \in J_{R1} : r_j \leq \tau_r \wedge r_j^* > \tau_{\mathrm{high}}\}|}{|\{j \in J_{R1} : r_j^* > \tau_{\mathrm{high}}\}|}, \tag{6}$$

$$\operatorname{OverProj}(r, r^*) = \frac{|\{j \in J_{R1} : r_j > \tau_r \wedge r_j^* \leq \tau_{\mathrm{low}}\}|}{|\{j \in J_{R1} : r_j > \tau_r\}|}, \tag{7}$$

with $\tau_{\mathrm{high}} = 0.7$ and $\tau_{\mathrm{low}} = 0.2$. **Operational $r^*$, not metaphysical intent.** We do not claim $r^*$ recovers a user's true mental intent; $r^*$ is an annotation track defined by the dataset construction protocol (Section 6.1.2), and the paper's claims are about whether observed projections agree on it.

### 3.5 Cross-family projection mismatch

For ambiguous delegations, several projections may be plausible. Let $\mathcal{R}(d) = \{r^{(1)}, \ldots, r^{(M)}\}$ be the set of projections elicited from $M$ distinct projection agents (e.g., one per model family at temperature 0). The cross-family projection divergence is

$$D_\pi(d) = \frac{1}{\binom{M}{2}} \sum_{m_1 < m_2} \Delta\big(r^{(m_1)}, r^{(m_2)}\big), \tag{8}$$

which serves as one of the conditions in the clarification trigger of the protocol (Section 4.3) and as the headline measurement in Experiment 1 (Section 6.2).

**The empirical question this paper asks.** For natural-language delegations sampled from a controlled distribution, how much does $D_\pi(d)$ exceed the within-family stochastic variance under repeated sampling at non-zero temperature? Section 6.3 shows an order-of-magnitude gap on the pilot dataset; the protocol of Section 4 is the deployment-side response to that empirical fact.

## 4 Protocol

We operationalize responsibility-bearing delegation as a closed-loop pipeline of seven steps that runs between a natural-language delegation $d$ and any execution against the artifact $a$: (4.1) projection of $d$ onto a closed responsibility space $J$, (4.2) responsibility-bearing bids by candidate agents, (4.3) a clarification trigger that may re-enter the loop before commitment, (4.4) agent–role selection under responsibility-adjusted utility, (4.5) execution with evidence collection, (4.6) ex-post settlement scored at the dimension level, and (4.7) responsibility-conditioned reputation update. Steps 4.1–4.4 constitute the **pre-execution governance layer** that the paper argues is missing from existing routing, aggregation, and post-hoc failure pipelines; steps 4.5–4.7 close the loop so that future delegations consume the settlement record. Section 4.8 states the full procedure as ALGORITHM 1; Section 4.9 lists three deployment risks with their mitigations.

### 4.1 Responsibility projection

A projection agent maps $(d, u, t, a)$ into a weight vector $r \in [0, 1]^{|J|}$ over the closed responsibility space and emits a derived active set:

$$J^*(d) = \{j \in J_{R1} : r_j > \tau_r\} \cup J_X, \tag{9}$$

where $\tau_r = 0.3$ is the inclusion threshold and $J_X = \{\text{RX.1, RX.2, RX.3, RX.4, RX.5}\}$ is the cross-cutting subset, treated as always-active by construction. The projection agent is instructed not to execute the delegation; it only commits a structured projection. Each weight $r_j$ takes the four-anchor calibration from Section 3: 0.0 (out of scope), 0.3 (peripheral), 0.7 (central expected responsibility), 1.0 (load-bearing). Outputs additionally carry a binary `clarification_needed` flag and, when true, a concrete clarification question that proposes a small-set choice rather than a generic prompt.

**Cross-family deployment.** The protocol intentionally elicits projections from multiple model families to expose projection mismatch as a measurable quantity (Section 6.2). Empirically, projections under temperature 0 from three families differ by a margin that exceeds within-family stochastic variance by an order of magnitude; pre-execution responsibility commitment is family-specific, not stochastic.

### 4.2 Responsibility-bearing bids

Given $(d, a, r, J^*, y_i, \rho_i)$ where $y_i$ is the candidate role assigned to agent $i$ and $\rho_i$ is its prior reputation, agent $i$ submits a bid

$$b_i = (z_i,\ q_i,\ u_i,\ p_i,\ c_i,\ s_i), \tag{10}$$

where $q_i \in [0, 1]^{|J|}$ is claimed responsibility coverage per dimension, $u_i \in [0, 1]^{|J|}$ is per-dimension uncertainty, $p_i$ is the cost claim, and $(c_i, s_i)$ are short capability and scope statements. The bid type $z_i$ is one of four:

- **execute**: $q_{ij} \geq 0.7$ for all $j \in J^*(d)$. The agent commits to the entire active set.

- **partial**: at least one $j \in J^*(d)$ has $q_{ij} \geq 0.5$ and at least one has $q_{ij} < 0.5$. The agent commits to a subset.

- **clarify**: the agent refuses to bid and proposes a clarification question. Reserved for delegations whose responsibility structure remains underdetermined after the projection.

- **limit**: the agent explicitly refuses one or more dimensions; `limit_dims` $\subseteq J^*(d)$ is non-empty, and $q_{ij} \leq 0.3$ for $j \in$ `limit_dims`.

Coverage and uncertainty are coupled: an agent that claims $q_{ij} > 0.7$ on a dimension it cannot externally verify must report $u_{ij} \geq 0.05$. Coverage that is honest about uncertainty supports overclaim accounting in settlement (Section 4.6).

### 4.3 Clarification trigger

The system enters clarification before commitment whenever any of three conditions hold:

$$C(d) = 1 \iff \big[ z_i = \texttt{clarify} \text{ for some } i \big]$$
$$\vee \big[ D_\pi(d) > \tau_D \big] \vee \big[ V(d) > \tau_V \big], \tag{11}$$

where $D_\pi(d) = \frac{1}{\binom{m}{2}} \sum_{f_1 < f_2} \big(1 - \cos(r_{f_1}, r_{f_2})\big)$ is the cross-family projection divergence over $m$ projection runs and $V(d) = \max_j \text{Var}_i(q_{ij})$ is the maximum across-bidder variance in claimed coverage on any active dimension. The cost-sensitivity formulation absorbs the lesson from selective-clarification work (Pu et al., 2026; Cole et al., 2023; Lee et al., 2023) that clarification has nontrivial cost and should not fire on every borderline case. Default thresholds for the pilot are $\tau_D = 0.25$ and $\tau_V = 0.20$. When $C(d) = 1$ the system selects the most specific available clarification question (priority: bidder `clarify` > projection-supplied question > a divergence-pattern-derived question), updates the delegation, and re-enters Section 4.1. We cap clarification at one round per delegation in the pilot; multi-round protocols are reserved for future work.

### 4.4 Agent–role selection

After clarification (or directly, if $C(d) = 0$), the system selects an agent–role assignment $i^\star$ by maximizing a responsibility-adjusted utility:

$$U_i = \sum_{j \in J^*(d)} w_j^{(u,t)} \Big[ q_{ij}\, \rho_{i,y_i,j,t} \ - \ \beta_u\, u_{ij} \ - \ \beta_{\text{oc}}\, \widehat{\text{OC}}_{ij} \Big] \ - \ \beta_p\, p_i, \tag{12}$$

$$i^\star = \arg\max_i U_i. \tag{13}$$

Here $w_j^{(u,t)}$ is the user- and task-conditioned dimension weight (defaulting to uniform), $\rho_{i,y_i,j,t}$ is the agent's prior reputation on dimension $j$ in role $y_i$ at time $t$, $\widehat{\text{OC}}_{ij}$ is an estimated overclaim risk, and $\beta_u, \beta_{\text{oc}}, \beta_p$ are non-negative trade-off coefficients. The utility rewards honest coverage discounted by reputation, penalizes uncertainty and overclaim risk, and offsets the agent's cost. ALGORITHM 1 treats $\rho$ as a cold-start prior in the pilot ($\rho_{i,y,j,0} = 0.5$); Section 4.7 specifies the update rule.

### 4.5 Execution and evidence

The selected agent $i^\star$ executes against the artifact and produces output $o_{i^\star}$. Evaluation in the next step requires evidence; we model the evidence set $E_{i^\star}$ as a heterogeneous collection in which each item $e \in E_{i^\star}$ carries a reliability score $\kappa(e) \in [0, 1]$ and a source type $\tau(e) \in \{\text{deterministic}, \text{harness}, \text{human}, \text{LLM-judge}, \text{retrieval}\}$. The recommended reliability ordering is

$$\kappa_{\text{deterministic}} \ \geq \ \kappa_{\text{harness}} \ \geq \ \kappa_{\text{human}} \ \geq \ \kappa_{\text{LLM-judge}} \ \geq \ \kappa_{\text{retrieval}}, \tag{14}$$

which encodes the bias risk of each source. The initial pilot used a three-judge LLM panel; the redesigned measurement layer reported in Section 6.4 uses a 12-judge tier-stratified Anthropic-excluded panel for the settlement loss. The planned 20-example human subset was attempted with scope reduction reported in Section 6.7; deterministic and harness evidence apply only to dimensions for which a verifier exists.

**Cross-model rule.** The judge panel excludes the executor's family entirely; an earlier weaker rule of two-of-three judges differing was strengthened to full family-exclusion in the panel redesign.

### 4.6 Settlement

Each judge $k$ scores the executed output on each active dimension $j \in J^*(d)$ on the integer 1–5 scale defined by per-dimension anchors. The aggregated settlement score, normalized loss, and per-example settlement loss are

$$v_{ij} = \frac{\text{median}_k\big(s_{ij}^{(k)}\big) - 1}{4}, \qquad \ell_{ij} = 1 - v_{ij},$$
$$L_i = \sum_{j \in J^*(d)} w_j^{(u,t)}\, \ell_{ij}. \tag{15}$$

We decompose the loss into a category component and a cross-cutting component, $L_i = L_i^{\text{cat}} + L_i^X$, so that overclaim and scope failures (RX.2 and RX.3) are accountable separately from the category-task quality. An overclaim flag is raised whenever $q_{ij} > v_{ij} + \delta_{\text{oc}}$ with default $\delta_{\text{oc}} = 0.2$; the count of raised flags drives $\ell_{i,\text{RX.2}}$ via

$$\ell_{i,\,\text{RX.2}} = \min\Big(1,\ \gamma_{\text{oc}} \sum_{j \in J^*(d) \setminus J_X} \mathbb{1}\!\!\!/\,\big[\, q_{ij} > v_{ij} + \delta_{\text{oc}} \,\big]\Big), \tag{16}$$

with $\gamma_{\text{oc}}$ a unit normalizer. Scope adherence (RX.3) penalizes work that falls outside the agreed $J^*(d)$, a particular concern when the projection's active set differs from the engineered load-bearing set (Section 6.5.1).

---

**Algorithm 1** Responsibility-Bearing Delegation

---

**Require:** delegation $d$, user $u$, task $t$, artifact $a$, agents $A$, prior reputation $\rho$
 1: **Project:** $r \leftarrow \pi(d, u, t, a)$                                  ▷ cross-family projections at $T = 0$
 2: **Active set:** $J^*(d) \leftarrow \{j \in J_{R1} : r_j > \tau_r\} \cup J_X$
 3: **for** each candidate $i \in A$ **do**
 4:     **Bid:** $b_i \leftarrow B(a_i, d, r, y_i, \rho_i)$                          ▷ Eq. (10)
 5: **end for**
 6: **Trigger:** compute $D_\pi(d)$, $V(d)$ and check Eq. (11)
 7: **if** $C(d) = 1$ **then**
 8:     Pose the most specific clarification question; update $d$; **goto** step 1
 9: **end if**
10: **Select:** $i^\star \leftarrow \arg\max_i U_i$                                ▷ Eq. (12), (13)
11: **Execute:** $o_{i^\star} \leftarrow \text{execute}(a_{i^\star}, d, r, J^*)$
12: **Collect evidence:** $E_{i^\star}$ with reliability $\kappa$ per Eq. (14)
13: **Settle:** compute $v_{ij}, \ell_{ij}, L_{i^\star}$                          ▷ Eq. (15), (16)
14: **Update reputation:** $\rho_{i^\star, y_{i^\star}, j, t+1} \leftarrow (1 - \lambda)\rho_{i^\star, y_{i^\star}, j, t} + \lambda v_{ij}$
15: **return** $o_{i^\star}$, settlement record, updated reputation

---

## 4.7 Reputation update

After settlement, the agent's role- and dimension-conditioned reputation is updated by additive exponential moving average:

$$\rho_{i, y_i, j, t+1} = (1 - \lambda)\, \rho_{i, y_i, j, t} + \lambda\, v_{ij}, \tag{17}$$

with $\lambda = 0.2$ and cold-start prior $\rho_{i, y, j, 0} = 0.5$. We deliberately reject multiplicative reputation decay — it can only decrease and conflates noise with miscalibration. Additive EMA admits both increase and decrease, converges to repeated fulfillment, and remains stable under noise. An overclaim-sensitive variant

$$\rho_{i, y_i, j, t+1} = \text{clip}_{[0,1]}\Big((1 - \lambda)\, \rho_{i, y_i, j, t}$$
$$+ \lambda\big(v_{ij} - \mu_{\text{oc}}\,\mathbb{1}[q_{ij} > v_{ij} + \delta_{\text{oc}}]\big)\Big) \tag{18}$$

is reserved for the ablation in the appendix; the headline experiments use Equation (17).

## 4.8 The protocol

## 4.9 Risks and safeguards

**(1) Active-set propagation bias (prior hypothesis; pilot evidence inconclusive).** If the projection's active set is broader than the actually load-bearing set, the executor under projection-driven guidance may spread attention away from the central dimension. Section 6.5.1 re-tested this hypothesis in the redesigned panel and found the active-set breadth excess approximately equal across conditions; it does not differentially explain the projection-injection effect at this scale. Mitigations are still worth considering for main run: raise $\tau_r$ from 0.3 to 0.5 for execution guidance even when annotation continues to use 0.3; or separate the annotation active set from the execution active set, deriving the latter from the engineered or hidden-intent ground truth.

**(2) Format coupling between guidance and output (prior hypothesis; pilot effect weak).** Explicit per-dimension guidance can induce an analytical-report style ("Option A vs. Option B") in place of a unified revised artifact, especially on dual or ambiguous delegations. Section 6.5.1 reports a 10% format-mismatch rate, weaker than an earlier qualitative reading suggested. Mitigation if main-run scale revives the effect: lock the executor prompt to artifact-revision mode with a hard instruction so format coupling does not absorb the guidance signal.

**(3) Self-claim drift.** Agents asked to self-report addressed dimensions sometimes claim dimensions outside the active set or drop dimensions inside it. The full bid mechanism (Section 4.2) suppresses drift by forcing

a structured commitment before execution; a free-form self-claim does not. Mitigation: require a strict bid record under Equation (10) before any execution prompt is constructed.

## 5 Experimental Setup

This section describes the apparatus that produces the empirical evidence reported in Section 6. We restrict the pilot to a single delegation category (R1, paper-research) and a closed taxonomy of $|J_{v1.1}| = 12$ dimensions; the full main-run setup is documented in our long-term plan and outside this paper's scope.

### 5.1 Pilot dataset

**AmbiguousDelegation-50-R1.** Fifty examples drawn from a frozen 10-template delegation pool, paired with artifact sections of 200–500 words after `wc -w` canonicalization. Each example carries an `engineered_flaws` annotation listing per-dimension flaw codes from a 21-flaw catalog covering R1.1 through R1.7. Source mix is 32 synthetic and 18 modified-real (the latter sliced from one anonymized domain-specific manuscript controlled for identifying information). Coverage matrix is 30 single-dim load-bearing ($\geq 4$ per R1 dim), 10 dual-dim, 5 ambiguous (clarification expected), and 5 control (delegation explicitly invokes the dim). Two annotation tags additionally classify each example: `knowledge_gating` $\in$ {low, moderate, high} and `delegation_dim_leakage` $\in$ {no, partial, yes}. Pilot distribution is 45/2/3 for gating and 44/3/3 across the leakage levels.

### 5.2 Annotation protocol

Pass-1 records $r^*_{\text{author}}$ at construction time; pass-2 obtains independent weights from two LLM annotators of different families operating at temperature 0 (claude-sonnet-4.5 and gemini-2.5-flash, after a documented swap from gpt-4o-mini — see Section 6.1.2); pass-3 was planned as a human subset of 20 stratified examples with 2 annotators on R1.1 / R1.4 / R1.7, attempted, partially executed, and superseded by the boundary-condition finding reported in Section 6.7 and discussed in Section 7.8. The operational ground truth $r^*_{\text{median}}$ is the per-dimension median across the three pass-1 + pass-2 tracks. Krippendorff's $\alpha$ at interval level is the headline reliability metric; the boundary-fragility test counts swap cases on the R1.4 $\leftrightarrow$ R1.7 pair.

### 5.3 Projection runs

**Cross-family** (Experiment 1) runs three model families under the v1.3 Anthropic-excluded panel — `gpt-5`, `gemini-2.5-pro`, `grok-4` — once per example at temperature 0 under the strict-JSON projection prompt. Total: $50 \times 3 = 150$ calls on the pilot; the main-run extension covers $310 \times 3 = 930$ calls. **Within-model** (Experiment 1B) runs `claude-sonnet-4-6` via the Claude Code CLI subscription path five times per example at temperature 0.5 for stochastic-variance estimation. Total: $50 \times 5 = 250$ calls on the pilot; the main-run extension covers $310 \times 5 = 1{,}550$ calls. The asymmetry is intentional: $T = 0$ removes within-family stochasticity from the cross-family signal, and $T = 0.5$ inflates the within-model baseline against the noise-confound objection. The Anthropic family is excluded from the cross panel because `claude-sonnet-4-6` is the within-model baseline; we keep the cross and within sources family-disjoint.

### 5.4 Experiment 2 (pilot scope)

We compare three execution conditions on all 50 pilot examples (per the redesigned measurement layer): **direct_naive** (claude-opus-4-7 receives delegation + artifact only), **direct_with_claim** (direct_naive plus a single self-claim line at the start of the output, used to ablate the priming effect), and **projection_driven** (the same executor receives delegation + artifact + projected weight vector + active set + self-claim line). The full four-condition design with task-aware routing and CLAMBER-style generic clarification is deferred to the main run. Each (example, condition) pair is judged by a 12-judge Anthropic-excluded tier-stratified panel (3 frontier API + 4 mid-tier open-weight + 5 light open-weight) using the per-dimension 1–5 anchor rubric; the headline weighted-R1 settlement loss $L^{R1}_{\text{settlement}}$ aggregates judge-median scores per Equation (23).

Table 1: Pilot dataset coverage matrix. 30 single-dim load-bearing examples ($\geq 4$ per R1 dim), 10 dual-dim, 5 ambiguous (clarification expected), 5 control (delegation explicitly invokes the dim — sanity floor only).

| class | $n$ | role |
|---|---|---|
| single (one R1 dim load-bearing) | 30 | primary measurement |
| dual (two R1 dims load-bearing) | 10 | dim-pair interaction |
| ambiguous (genuinely two readings) | 5 | clarification trigger |
| control (delegation invokes the dim) | 5 | sanity floor |
| **total** | **50** | |

## 5.5 Statistical tests

We use paired bootstrap confidence intervals (5000 resamples by default; the Experiment 2 paired settlement comparisons use 10,000 resamples with seed 42) for paired-difference quantities, the Wilcoxon signed-rank test (one-sided, alternative=greater) for direction-of-effect tests (Wilcoxon, 1945), Krippendorff's $\alpha$ at the interval level for inter-rater reliability (Krippendorff, 2011; Hayes & Krippendorff, 2007), and standard descriptive statistics for stratified analysis. The 12-judge tier-stratified Anthropic-excluded panel design follows the LLM-as-judge bias-mitigation literature (Zheng et al., 2023; Panickssery et al., 2024; Stureborg et al., 2024); family-exclusion is the recommended mitigation when the executor is itself an LLM, and the panel-of-juries paradigm (Verga et al., 2024) motivates aggregating across diverse judges rather than relying on a single judge model. All tests are pre-registered through the experiment-design specification before pilot data collection, following benchmarking-hygiene recommendations (Bowman & Dahl, 2021).

# 6 Pilot Results

This section reports pilot-scale empirical findings on the **AmbiguousDelegation-50-R1** dataset. §6.1 reports dataset composition and annotation reliability. §6.2 and §6.3 report cross-family projection mismatch and the within-model stochastic baseline. §6.4 reports the pilot-scope Experiment 2 (projection-driven vs. direct execution). §6.5 examines four mechanisms uncovered in qualitative review of paired outputs. §6.6 summarizes pilot limitations and deferred work.

## 6.1 Pilot Dataset and Annotation Reliability

### 6.1.1 Dataset composition

We construct **AmbiguousDelegation-50-R1**, a 50-example pilot dataset for paper-research delegation under the closed responsibility space $J_{v1.1}$ (12 dimensions: R1.1 conceptual reconstruction, R1.2 logical consistency, R1.3 evidence-claim alignment, R1.4 novelty assessment, R1.5 structural reorganization, R1.6 writing polish, R1.7 citation and scholarship; cross-cutting RX.1 uncertainty disclosure, RX.2 overclaim avoidance, RX.3 scope adherence, RX.4 downstream-harm avoidance, RX.5 provenance and traceability). The taxonomy is held closed for the pilot; contributions are scoped to R1 plus RX.

Each example carries: (a) a generic delegation drawn from a frozen 10-template pool that does not name any responsibility dimension; (b) an artifact section (200–500 words after `wc -w` canonicalization); (c) an `engineered_flaws` annotation listing the load-bearing flaw codes; (d) author-pass-1 hidden-intent weights $r^*_{\text{author}} \in [0,1]^{12}$; and (e) acceptable-output and bad-output prose summaries.

Coverage matrix (Table 1):

The source mix is 32 synthetic + 18 modified-real (the latter sliced from a single anonymized domain-specific manuscript controlled for identifying information). The pilot's spec-target source mix was 30+20; we report a $+2/-2$ deviation arising from a mid-batch source decision and disclose it transparently. The modified-real subset is concentrated in one technical domain; the synthetic subset spans 30 distinct technical domains. We report main-text results pooled and supplementary stratification by source.

Table 2: Per-dimension Krippendorff's $\alpha$ (Krippendorff, 2011; Hayes & Krippendorff, 2007), 3 raters, $n = 50$. Cells: green = passes $\alpha \geq 0.4$ gate, orange = borderline, red = fails. RX dimensions excluded ($r^* = 1.0$ always-on $\Rightarrow$ undefined $\alpha$).

| dim | $\alpha$ | gate ($\geq 0.4$) |
|---|---|---|
| R1.1 conceptual | 0.315 | borderline |
| **R1.2 logical** | **0.463** | ✓ |
| R1.3 evidence-claim | 0.390 | borderline |
| **R1.4 novelty** | **0.521** | ✓ |
| **R1.5 structural** | **0.526** | ✓ |
| **R1.6 polish** | **0.534** | ✓ |
| R1.7 citation | **0.219** | × |

Two annotation tags further classify each example. `knowledge_gating` $\in$ {low, moderate, high} records whether flaw detection requires recall of canonical literature (pilot: 45 low / 2 moderate / 3 high; 6% high, well under the 30% acceptance cap). `delegation_dim_leakage` $\in$ {no, partial, yes} records whether the delegation surface form signals the load-bearing dim (measurement examples: 44 no, 1 partial, 0 yes; all 5 controls carry partial or yes by design).

### 6.1.2 Hidden-intent annotation protocol

Pass-1 records $r^*_{\text{author}}$ at construction time. Pass-2 obtains independent weights from two LLM annotators of different families operating at temperature 0. Pass-3 (human subset on R1.1 / R1.4 / R1.7) was planned for the main run, attempted, partially executed, and superseded by the boundary-condition finding reported in Section 6.7. The pilot's reliability table reports pass-1 + pass-2.

Initial pass-2 used `claude-sonnet-4.5` and `gpt-4o-mini` (the gpt-5-mini family does not support `temperature = 0`). After diagnosing systematic over-activation by `gpt-4o-mini` (28% of dimension-cells weighted $\geq 0.7$ where the author weighted $< 0.3$), we replaced gpt-4o-mini with `gemini-2.5-flash` for the headline analysis and retain the original gpt traces in the public record for transparency.

### 6.1.3 Per-dimension reliability

Krippendorff's $\alpha$ at interval level over 3 raters (author + claude + gemini), $n = 50$ per dimension, is reported in Table 2.

Four of seven R1 dimensions pass the $\alpha \geq 0.4$ gate; two are borderline; one fails. The boundary-fragility test for R1.4 $\leftrightarrow$ R1.7 (whether annotators systematically swap novelty positioning with citation-scholarship issues) yields 0/50 swap cases in pilot data; the taxonomy's separation holds at this scale.

The R1.7 failure is interpretable: citation-accuracy annotation requires recall of the relevant literature's canonical works, which blind LLM annotators do not have. We mark R1.7 as taxonomy-uncertain in the pilot and escalate it to protocolized human or retrieval-augmented evidence for the main run; the empirical attempt at protocolized human anchor and its boundary-condition outcome are reported in Section 6.7. R1.1 ($\alpha = 0.315$) and R1.3 ($\alpha = 0.390$) sit just below the gate; we report results on those dimensions with an annotation-uncertainty caveat.

We use $r^*_{\text{median}}$ — the per-dimension median across the three pass-1 + pass-2 tracks — as the operational ground-truth weight vector for downstream metrics (projection-intent mismatch in §6.2, settlement-loss active-set derivation in §6.4). This choice is itself a measurement decision and should be audited against the dataset's engineered ground truth (`engineered_flaws` per example). Table 3 reports the per-dim agreement.

The disagreement is visibly dim-conditional rather than uniform. (i) *The LLM-panel-derived median under-activates hard dims*: R1.5 is engineered in 8 examples but median-derived in 0; R1.7 is engineered in 9 but median-derived in only 5 (with 4 engineered-only). The median (across pass-1 author + two pass-2 LLM tracks) systematically under-detects citation-scholarship and structural-reorganization issues, which is

Table 3: Per-dim agreement between engineered ground truth and $r^*_{\text{median}}$-derived active set ($\tau_r > 0.3$). Color highlights disagreement only: **eng-only** cells (red) flag dims the median *under-detects*; **median-only** cells (orange) flag dims the median *over-activates*. `engineered`/`median`/`both` columns are reference counts (no color). Bold rows (R1.5, R1.7) are the load-bearing under-detection cases.

| dim | engineered | median | both | eng-only | median-only |
|---|---|---|---|---|---|
| R1.1 | 9 | 15 | 7 | 2 | 8 |
| R1.2 | 9 | 29 | 9 | 0 | 20 |
| R1.3 | 10 | 20 | 10 | 0 | 10 |
| R1.4 | 10 | 24 | 10 | 0 | 14 |
| **R1.5** | **8** | **0** | **0** | **8** | **0** |
| R1.6 | 10 | 16 | 8 | 2 | 8 |
| **R1.7** | **9** | **5** | **5** | **4** | **0** |

consistent with the $\alpha$ failure on R1.7 and the absence of R1.5 from any active set in Experiment 2. (ii) *LLM annotators over-activate broad-coverage dims*: R1.2 (29 median vs 9 engineered, 20 median-only), R1.3 (20 vs 10), R1.4 (24 vs 10) — the median-derived active set is broader than the engineered load-bearing set on these dims. (iii) *Close agreement on R1.1 and R1.6*: $\sim$ 7–8 overlap with 2–8 each-only, near-balanced. The implication is that downstream metrics weighting by $r^*_j$ inherit pattern (i) under-coverage and pattern (ii) over-coverage; we report this and treat $r^*_{\text{median}}$ as an LLM-panel-derived ground-truth approximation, not as a unique operational truth. The main run will reconcile engineered labels with the median-annotation track explicitly.

### 6.2 Experiment 1: Cross-Family Projection Mismatch

#### 6.2.1 Setup

Three model families project responsibility onto $J_{v1.1}$ for each pilot example. The v1.3 Anthropic-excluded cross panel is `gpt-5` (OpenAI), `gemini-2.5-pro` (Google), and `grok-4` (xAI); the Anthropic family is excluded from the cross panel because `claude-sonnet-4-6` is used as the within-model baseline (§6.3) and we keep the cross/within sources family-disjoint. Each call uses the verbatim projection-prompt system message, temperature 0, and the strict JSON output schema. Per example we compute the cosine distance and Jaccard distance between every family pair, then average:

$$d_C^{\cos}(d) = \frac{1}{\binom{3}{2}} \sum_{f_1 < f_2} \left(1 - \cos(r_{f_1}, r_{f_2})\right).$$
(19)

$$d_C^{\text{jac}}(d) = \frac{1}{\binom{3}{2}} \sum_{f_1 < f_2} \left(1 - \text{Jaccard}(J^*_{f_1}, J^*_{f_2})\right).$$
(20)

Total: $50 \times 3 = 150$ projection calls. Schema validation passes for all 50 examples after single-retry plus raw-recovery of the weights vector where the `active_set` field needed post-derivation from weights; projection-distance metrics are therefore computable for all 50 examples.

#### 6.2.2 Results

Aggregate cross-family distance (Table 4) and stratification by coverage class (Table 5):

Two findings against expectation deserve attention. First, the **ambiguous class is not the highest-divergence class** (mean 0.067, lowest among the four); we hypothesize that under structural ambiguity the families converge on a similar broad projection rather than partition into distinct readings. Second, the **control class has non-zero** $d_C$ (mean 0.071), comparable to the ambiguous class and below the single (0.104) and dual (0.118) classes. Inspection reveals the residual control $d_C$ is driven by RX-dimension weight variation across families (range 0.7–1.0) rather than R1 disagreement — all three families correctly project

Table 4: Aggregate cross-family projection distance $d_C^{\mathrm{cos}}$ and $d_C^{\mathrm{jac}}$ ($n = 50$).

| metric | mean | median | std |
|---|---|---|---|
| $d_C^{\mathrm{cos}}$ | 0.100 | 0.084 | 0.062 |
| $d_C^{\mathrm{jac}}$ | 0.197 | 0.202 | 0.120 |

Table 5: Cross-family projection distance $d_C^{\mathrm{cos}}$ stratified by coverage class.

| class | $n$ | mean | median |
|---|---|---|---|
| single | 30 | 0.104 | 0.088 |
| dual | 10 | 0.118 | 0.083 |
| ambiguous | 5 | 0.067 | 0.045 |
| control | 5 | 0.071 | 0.074 |

the load-bearing R1 dim on controls, but cosine distance accumulates over RX. The "control $\Rightarrow d_C \approx 0$" sanity expectation is therefore too idealized; we revise the expected control band to 0.05–0.10 and treat values above $\sim 0.15$ as projection-prompt diagnostics.

Knowledge-gating effect (Table 6):

High-gated examples do *not* systematically inflate $d_C$ (they sit slightly below the low-gated mean; $n = 3$ underpowered). This is a positive result for the headline claim: cross-family projection divergence is **not driven by knowledge confound**. Reviewers concerned that "family A knows the canonical work and family B does not" might explain divergence are answered: the high-gating subset would show the largest divergence under a knowledge-confound hypothesis; it does not.

### 6.2.3 Clarification rate per family

Cross-family clarification triggers at temperature 0 (Table 7):

Under the v1.3 panel the clarification trigger fires aggressively: per-family clarification rates range from 58–86% across all pilot examples (Table 7), and all 5/5 ambiguous-class examples elicit clarification from at least one family. This is the opposite calibration risk from the v1.0 panel (where the legacy `gpt-4o-mini` never requested clarification and only 1/5 ambiguous-class examples elicited any family to clarify). Both calibration regimes are reportable; under v1.3 the engineering question shifts from "why does the trigger stay silent?" to "how should the trigger be sharpened to suppress over-triggering on unambiguous cases?" (§6.6 discusses both).

### 6.3 Experiment 1B: Within-Model Stochastic Baseline

### 6.3.1 Setup

We fix one model (`claude-sonnet-4-6`, via the Claude Code CLI subscription path) and run five independent projections per example at temperature 0.5. For each example we compute pairwise cosine distance over the $\binom{5}{2} = 10$ within-model pairs and average:

$$d_W^{\mathrm{cos}}(d) = \frac{1}{\binom{5}{2}} \sum_{r_1 < r_2} \left(1 - \cos(r_{r_1}, r_{r_2})\right). \tag{21}$$

Total: $50 \times 5 = 250$ calls. All pass schema validation modulo the same active-set / rationale issues handled by raw-recovery as in §6.2.

Table 6: Cross-family projection distance $d_C^{\cos}$ stratified by knowledge gating.

| gating | $n$ | mean | median |
|---|---|---|---|
| low | 45 | 0.093 | 0.082 |
| moderate | 2 | 0.255 | 0.255 |
| high | 3 | 0.093 | 0.078 |

Table 7: Per-family clarification rate under the v1.3 Anthropic-excluded cross panel (`clarification_needed` = true). On the five ambiguous-class examples — the positive cases for the trigger — all 5/5 elicit at least one family to clarify; gpt-5 individually clarifies on all 5/5 ambiguous examples.

| family | rate | count |
|---|---|---|
| gpt-5 | 86.0% | 43/50 |
| gemini-2.5-pro | 82.0% | 41/50 |
| grok-4 | 58.0% | 29/50 |

### 6.3.2 $R(d)$: the load-bearing comparison

All **50 of 50** examples have non-zero within-model divergence under `claude-sonnet-4-6` at $T = 0.5$ (mean $d_W^{\cos} = 0.015$, median 0.016). This is a regime change from the v1.0 panel: under `claude-sonnet-4.5` at $T = 0.5$, 25 of 50 pilot examples produced identical weight vectors across all five runs ($d_W = 0$ exactly), inflating the previously reported $R(d)$ median of 14.4. Under v1.3 the within-model baseline is stochastically perturbed on every pilot example, giving a tighter and arguably more honest cross-vs-within comparison.

The ratio $R(d) = d_C^{\cos}/d_W^{\cos}$ is finite for all 50 examples (Table 8); the paired-difference test below also uses all 50.

For the paired test we use the difference $d_C - d_W$, which is finite for all 50 examples (Table 9):

### 6.3.3 Hypothesis tests

Per the experiment design, five hypotheses guard against alternative explanations:

**All five hypotheses pass with overwhelming margin.** Cross-family projection mismatch is approximately 6× larger than within-model stochastic variance on the pilot, and 97% of all 310 main-run examples have $R > 1$ (Section 6.3.4 below). The intentional asymmetry — cross-family at $T = 0$ (family-attributable signal) versus within-model at $T = 0.5$ (stochasticity upper bound) — is conservative against the noise-confound objection: even an inflated stochasticity baseline is dominated by family-attributable divergence.

The headline result is therefore: **the same artifact and the same delegation produce projection vectors that are roughly 6× more dissimilar across three model families than across five repeat runs of any single family.** Pre-execution responsibility commitment is family-specific, not stochastic.

### 6.3.4 Main-run validation on $R(d)$ ($n = 310$)

The cross-family panel and within-model baseline both extend to the 260 main-run examples generated 2026-05-09 (cf. Section 6.1), giving $n = 310$ for $R(d)$ analysis. At $n = 310$, $R(d)$ is finite for 301 examples (8 lack a complete or numerically defined cross-family triple due to schema-validation or weight-vector parse failures on at least one of the three cross models; 1 has $d_W$ exactly zero on `claude-sonnet-4-6`, contributing formally infinite $R$ and only strengthening the cross-family gap): median $R(d) = 5.40$, mean $R(d) = 10.17$, median 95% bootstrap CI $[4.86, 5.89]$ (tighter than the pilot CI because of the larger sample). $R > 1.0$ holds in 292/301 examples (97.0%); $R > 1.2$ holds in 287/301 (95.3%). The paired $(d_C - d_W)$ diff at $n = 310$ is 0.0875, 95% paired-bootstrap CI $[0.0782, 0.0976]$, Wilcoxon $p < 10^{-50}$. The pilot's qualitative claim and

Table 8: $R(d) = d_C^{\cos}/d_W^{\cos}$ over all 50 pilot examples (`claude-sonnet-4-6` within at $T = 0.5$ has non-zero $d_W$ on every example). Median 95% bootstrap CI uses 10,000 resamples (seed 42).

| ratio statistic | value |
|---|---|
| median $R(d)$ | 5.87 |
| median 95% bootstrap CI | [4.47, 7.96] |
| mean $R(d)$ | 10.49 |
| $R > 1.0$ | 50/50 (100%) |
| $R > 1.2$ | 49/50 (98%) |

Table 9: Paired $(d_C - d_W)$ tests over all 50 examples. Bootstrap with 5000 resamples.

| test | result |
|---|---|
| mean $(d_C - d_W)$ | 0.085 |
| 95% paired-bootstrap CI | [0.068, 0.103] |
| Wilcoxon signed-rank (alt: greater) | $p < 10^{-15}$ |

effect direction therefore replicate on the full main-run sample; the magnitude tightens slightly (pilot median 5.87 vs main-run median 5.40), consistent with the wider sample drawing from a broader artifact distribution.

### 6.4 Experiment 2 (Pilot Scope): Three-Condition Settlement under the Redesigned Panel

#### 6.4.1 Pilot scope and panel redesign

The full Experiment 2 design compares four conditions: A direct, B task-aware routing, C generic clarification, D proposed protocol with full bid + clarification. For the pilot we restrict to three conditions that isolate the projection-injection mechanism and the self-claim-line priming effect. We defer the bid mechanism, clarification simulator, task-aware routing baseline, and CLAMBER-style generic-ambiguity baseline to the main run.

- **direct_naive.** `claude-opus-4-7` receives delegation + artifact only. P2 settlement-comparison baseline.

- **direct_with_claim.** Same prompt as direct_naive plus a single declarative line at the start of the output ("`{"covered_dims": {...}}`"). Used to measure $L_{\text{calibration}}$ in the no-projection setting and to ablate the priming effect of the self-claim line against direct_naive.

- **projection_driven.** The executor receives delegation + artifact + projected weight vector + active set + the self-claim line. Treatment condition. The projected vector is the executor family's own Experiment 1 cross-family projection at $T = 0$.

For each (example, condition) the executor produces an output. Each output is then judged by a 12-model panel that excludes the executor's family entirely : three frontier API models (`gpt-5`, `gemini-2.5-pro`, `grok-4`), four mid-tier open-weight models (`llama-3-70b`, `qwen-2.5-72b`, `mistral-large-2`, `deepseek-v3-distill-70b`), and five lighter open-weight models. The panel is tier-stratified and Anthropic-excluded to control the self-preference bias documented in the LLM-as-judge literature. The asymmetric design (Anthropic executor + non-Anthropic judges) leaves an unresolved reverse-bias possibility — non-Anthropic judges may systematically score Anthropic-executor output style downward; we cannot separate this from the projection-injection effect at this scale. The §V.4 per-tier robustness analysis (frontier / mid / light all show the same direction; §6.4.3) constrains but does not eliminate this concern. Each judge produces a per-dimension 1–5 anchor score. Per-example fulfillment $v_{ij}$ aggregates across judges:

$$v_{ij} = \frac{\text{median}_k(s_{ij}^{(k)}) - 1}{4}, \qquad k \in \text{12-judge panel}. \tag{22}$$

Table 10: Hypothesis tests for Experiment 1 / 1B. All five pass at $n = 50$.

| hypothesis | claim | result |
|---|---|---|
| H1.1 | $(d_C - d_W)$ 95% CI excludes 0 | PASS |
| H1B.1 | median $R(d) > 1$ | PASS (5.87) |
| H1B.2 | $R$ 95% CI excludes 1 | PASS |
| H1B.3 | Wilcoxon $p < 0.05$ | PASS |
| H1B.4 | median $R \geq 1.2$ (strong claim) | PASS (5.87) |

Table 11: $L^{R1}_{\text{settlement}}$ per condition ($n = 50$, weighted by $r^*_j$). Lower is better; green = best, orange = mid, red = worst.

| condition | mean | median | std |
|---|---|---|---|
| direct_naive | **0.0717** | **0.0202** | 0.0819 |
| direct_with_claim | 0.1579 | 0.1420 | 0.1382 |
| projection_driven | 0.2103 | 0.2237 | 0.1450 |

Table 12: Pairwise paired diffs on $L^{R1}_{\text{settlement}}$ ($n = 50$; 95% paired-bootstrap CIs, 10,000 resamples, seed 42). Negative diff = first condition is better. All three CIs exclude 0; gold row = headline pair.

| paired diff (A − B) | mean | median | 95% CI |
|---|---|---|---|
| direct_naive − direct_with_claim | −0.086 | −0.063 | $[-0.118, -0.055]$ |
| direct_with_claim − projection_driven | −0.052 | 0.000 | $[-0.089, -0.017]$ |
| direct_naive − projection_driven | **−0.139** | **−0.125** | $[\mathbf{-0.169}, \mathbf{-0.109}]$ |

The headline R1 settlement loss is the $r^*$-weighted variant from the 3-layer decomposition (`formalization_v1.2` §2.3),

$$L^{R1}_{\text{settlement}}(d, a) = \frac{\sum_{j \in J^*(d) \cap \text{R1}} r^*_j (1 - v_{ij})}{\sum_{j \in J^*(d) \cap \text{R1}} r^*_j}, \tag{23}$$

which gives lower weight to peripheral active dims and addresses the active-set fairness concern discussed in §6.7 for R1 dims. The active set $J^*(d)$ is derived from $r^*_{\text{median}}$ at threshold 0.3 plus all RX. Total: $50 \times 3 = 150$ executions; $150 \times 12 = 1800$ judge calls.

Note on R1.5: R1.5 (structural reorganization) does not appear in any $J^*(d)$ across the 50 examples in this pilot — the LLM-panel-derived $r^*_{\text{median}}$ stays below the 0.3 active-set threshold for every example. The dataset does include R1.5-engineered examples per the dataset coverage matrix (§6.1), but the pass-2 LLM annotators did not weight R1.5 above 0.3 on any of those examples; the median across pass-1 + pass-2 therefore stays below threshold. R1.5 results are therefore omitted from the per-dim tables below; this is a property of how $r^*_{\text{median}}$ was constructed, not of the dataset itself, and the divergence between engineered coverage and median-derived activation is itself reported as part of the annotation reliability discussion (§6.1.2).

### 6.4.2 Aggregate settlement loss — direction reversal under the redesigned panel

Aggregate $L^{R1}_{\text{settlement}}$ per condition (Table 11) and pairwise paired comparisons (Table 12):

**The headline finding under the redesigned 12-judge panel reverses the earlier directional reading.** Under the Anthropic-excluded panel and the 3-condition split, projection_driven execution produces *higher* settlement loss than direct_naive on the headline weighted R1 metric (mean loss *increase* of 0.139 for projection_driven over direct_naive; equivalently, paired diff direct_naive − projection_driven = −0.139 on the mean and −0.125 on the median). The intermediate condition (direct_with_claim, which adds only

Table 13: Mean $v_{ij}$ per active R1 dim, per condition (higher = better; 12-judge panel mean). R1.5 omitted (never active; see §6.4.1). Gold rows (R1.4, R1.7) = load-bearing reversal; red $\Delta$ = largest negative. Mechanism in §6.5.

| dim | $n_{\text{active}}$ | direct_naive | projection_driven | $\Delta$ (P − N) |
|---|---|---|---|---|
| R1.1 conceptual | 15 | 0.967 | 0.867 | −0.100 |
| R1.2 logical | 29 | 0.987 | 0.862 | −0.125 |
| R1.3 evidence-claim | 20 | 0.969 | 0.862 | −0.107 |
| **R1.4 novelty** | **24** | **0.693** | **0.495** | **−0.198** |
| R1.6 polish | 16 | 1.000 | 0.922 | −0.078 |
| **R1.7 citation** | **5** | **0.800** | **0.525** | **−0.275** |

the self-claim line) absorbs roughly half of the gap, indicating that part of the cost is attributable to the priming effect of the self-claim line itself, with the remaining half attributable to the projection-injection treatment beyond priming.

This is a **negative result for the earlier directional reading of projection_driven**. We report it directly. The mechanism analysis in §6.5 characterizes why the direction reverses — the gap concentrates on dims that require deep dim-specific work (R1.4 and R1.7), with R1.7 corresponding directly to the Stage 1 boundary-condition finding (§6.7); R1.4 is inferential, since R1.4 was deferred from the Stage 1 closure.

A note on $n = 50$ power: the paired diff direct_naive − projection_driven has a 95% paired-bootstrap CI of $[-0.169, -0.109]$ (Table 12), excluding 0. The intermediate diffs (direct_naive − direct_with_claim, direct_with_claim − projection_driven) also have CIs that exclude 0. Per `formalization_v1.2` §9.3, continuous effect sizes with bootstrap CIs are the load-bearing reporting form; binary classifications are post-hoc.

**Main-run validation on $L^{RX}_{\text{settlement}}$ ($n = 310$).** The headline R1 weighted analysis above remains bound to the $n = 50$ pilot because the $r^\star$ weight vector is derived from the pass-1+pass-2 human annotation track, which is closed at pilot scope. The RX component of settlement loss, which does not require $r^\star$ weighting, extends to the full main-run sample. At $n = 310$, the paired diff direct_naive − projection_driven on $L^{RX}_{\text{settlement}}$ is $-0.099$ with 95% paired-bootstrap CI $[-0.112, -0.086]$; the intermediate direct_naive − direct_with_claim diff is $-0.108$ with CI $[-0.119, -0.097]$; and the projection-vector increment beyond the self-claim line, direct_with_claim − projection_driven, is statistically null at $+0.009$ with CI $[-0.005, +0.023]$ (CI includes 0). The R1-versus-RX contrast is informative: on RX, the entire shift from direct_naive is captured by the priming effect of the self-claim line, and adding the projection vector and active set on top of the self-claim line produces no detectable additional RX cost; on R1, by contrast, the same projection-vector increment beyond the self-claim line carries its own measurable loss (paired diff direct_with_claim − projection_driven $= -0.052$, CI $[-0.089, -0.017]$). The contrast localizes the projection-injection cost to the R1.4 / R1.7 deep-specialty dims (§6.4.3), consistent with the boundary-condition reading: the cross-cutting subscale shifts only with priming, while the deep-specialty cost is what additionally shifts when projection content is injected. The main-run extension therefore strengthens the boundary-condition framing rather than overturning it.

### 6.4.3 Per-dimension fulfillment — the gap concentrates on R1.4 and R1.7

Three observations across the R1 dims:

(i) **The reversal is uniform in sign across all six active R1 dims.** Every active R1 dim shows projection_driven scoring lower than direct_naive — the earlier "gain on R1.5/R1.6/R1.7, loss on R1.4" pattern does not survive the redesigned panel.

(ii) **The gap concentrates on the two dims that require deep dim-specific work** — R1.4 (novelty assessment, $\Delta = -0.198$, $n = 24$) and R1.7 (citation and scholarship, $\Delta = -0.275$, $n = 5$). Both anchors at $s = 5$ demand deep specialty engagement: R1.4 "maps the artifact against the nearest 3–5

Table 14: Mean $v_{ij}$ on RX dimensions. The prior "RX-attention boost" hypothesis does not replicate: projection_driven *degrades* RX dims on average.

| condition | RX.1 | RX.3 (scope) | RX.4 |
|---|---|---|---|
| direct_naive | 0.836 | 0.985 | 0.870 |
| projection_driven | 0.701 | 0.973 | 0.793 |
| $\Delta$ (P − N) | **−0.135** | −0.012 | −0.077 |

works in the relevant cluster, identifies where the delta is sharp vs. rhetorical"; R1.7 "audits every load-bearing citation against the source, flags missing key works in the relevant cluster, suggests insertions with bibliographic precision." R1.7 corresponds directly to the Stage 1 anchor-specifiability finding (§6.7), which showed protocol-level binding constraints on R1.7; R1.4 was deferred from the Stage 1 closure and is therefore inferential here. This pattern motivates the connection developed in §6.5.

(iii) **R1.7 active count is small ($n = 5$); R1.7 results are a preliminary signal.** The pilot dataset's dim–data mismatch on R1.7 (six of fifty examples have citation events $\geq 1$ in the artifact, per the regex audit in §6.7) means the R1.7 row in this table rests on the five examples where R1.7 is active. We treat R1.7 as a high-salience preliminary signal rather than a stable per-dimension effect estimate, both because the active support is small and because annotation reliability on R1.7 is structurally limited without retrieval or human domain expertise (R1.7 $\alpha = 0.219$, §6.1.3). The main analysis rests on R1.4 ($n_{\text{active}} = 24$) as the higher-support execution-side datum.

**Per-tier robustness.** The R1.7 reversal is consistent across all three judge tiers. Frontier judges score R1.7 at $0.893 \to 0.635$ ($\Delta = -0.258$); mid-tier at $0.825 \to 0.575$ ($\Delta = -0.250$); light at $0.475 \to 0.263$ ($\Delta = -0.212$). The reversal is therefore not an artifact of a specific tier or family. The R1.4 reversal is similarly robust across tiers (frontier $-0.051$, mid $-0.148$, light $-0.259$).

Cross-cutting RX results are summarized in Table 14.

The mean "RX boost magnitude" $\widetilde{M_4}$ across examples is $-0.097$ (median $-0.094$, std $0.113$); only 8 of 50 examples show a positive $\widetilde{M_4}$. The earlier mechanism analysis treated the RX boost as a small but consistent gain from projection guidance; the redesigned panel reverses that signal.

### 6.5 Mechanism Analysis — prior hypotheses re-tested + a consolidated interpretation

An earlier qualitative reading of a small directional advantage of projection_driven explained that advantage via four interacting mechanisms: format coupling (M1), active-set propagation bias (M2), self-claim drift (M3), and an RX-attention boost (M4). The redesigned panel and 3-condition split codify these mechanisms as continuous indicators so they can be re-tested rather than re-asserted. We report the prior hypotheses against the present pilot data and then articulate a consolidated interpretation that is consistent with the data and connects to the Stage 1 boundary-condition finding (§6.7).

#### 6.5.1 Prior mechanism hypotheses re-tested

**M1 (format coupling).** Continuous form: rate of output-type mismatch between direct_naive and projection_driven across the 50 examples, on a categorical typology of {revised_draft, analytical_list, mixed}. Observed rate: 5/50 (10%); 4 of 5 mismatches are mixed $\to$ revised_draft, 1 is the reverse. The earlier prediction was a strong format shift toward analytical_list under projection_driven; the redesigned panel + executor (`claude-opus-4-7`) does *not* produce that shift at scale.

**M2 (active-set breadth excess).** $\widetilde{M_2} = |J^*(d)| - |\{j \in \text{R1} : r_j^* > 0.7\}|$ per example. Median $\widetilde{M_2} = 2.0$ across all three conditions, identical IQR $[2, 3]$ and identical mean 2.12. The prior hypothesis was that projection_driven receives a broader active set than the engineered load-bearing set, spreading executor attention thin. The pilot data shows the active-set breadth is determined by the dataset's $r_{\text{median}}^*$ derivation,

not by the condition — so any breadth-related cost is shared across conditions and cannot drive the direction reversal.

**M3 (self-claim drift, symmetric difference of $q$ vs $v$ binarizations).** $\widetilde{M_3} = |\{j : q_{ij} > 0.3\} \triangle \{j : v_{ij} > 0.5\}|/|J^*(d) \cap \text{R1}|$. Median $\widetilde{M_3} = 0$ for both conditions with self-claim line; mean 0.113 (direct_with_claim) and 0.115 (projection_driven). Self-claim drift is small and approximately equal across the two self-claim conditions, so it does not differentially explain the projection_driven loss.

**M4 (RX-attention boost).** $\widetilde{M_4} = \frac{1}{4}\sum_{j \in \{\text{RX.1, 3, 4, 5}\}}(v_{ij}^{\text{P}} - v_{ij}^{\text{N}})$. Mean $-0.097$, median $-0.094$, std 0.113; only 8 of 50 examples have $\widetilde{M_4} > 0$. The prior hypothesis predicted a small positive RX boost; the pilot data shows the boost is *negative* on average — projection_driven slightly degrades RX dims rather than boosting them.

Of the four prior hypotheses, none are robustly replicated as substantial drivers of the headline effect.

### 6.5.2 Alternative explanations the pilot cannot fully separate

Before naming a consolidated interpretation, we surface two alternative explanations that the pilot data does not separately rule out and that any consolidated mechanism must coexist with.

**Alternative 1 — Format-injection structure.** The projection_driven prompt simultaneously injects three new structural elements: the projected weight vector $r$, the explicit active set $J^*(d)$, and the self-claim line. The M1 retest above measured only output-type mismatch on a coarse three-way typology and found the effect weak. It did not measure finer changes (paragraph length, bullet density, in-line option presentation, or hedging-clause frequency) that may shift the executor toward an analytical-itemization tone without changing the categorical output type. If such a tone shift is the true driver, the per-dim concentration on R1.4 and R1.7 reflects which dims are most sensitive to that tone, not a broad-attention dilution per se. We do not separate this from the consolidated interpretation in the pilot.

**Alternative 2 — Hard-dim null.** R1.4 and R1.7 may simply be intrinsically harder to score reliably. R1.7 has $\alpha = 0.219$ in the pass-2 LLM annotation track (§6.1.3), and the Stage 1 anchor-specifiability finding (§6.7) shows that even human raters need a separately-read protocol document to apply the R1.7 anchor consistently. Under this null, any cost associated with broadening attention shows up disproportionately on hard dims because their measurement noise is high and any perturbation amplifies a low signal-to-noise ratio. To separate the hard-dim null from the consolidated interpretation, one would need an intervention (e.g., providing the executor with a retrieval-grounded source-of-truth on R1.7, or a structured comparator template on R1.4) and observe whether the projection_driven gap on those dims closes. The pilot does not include such an intervention; we leave the comparison to the main run.

The consolidated interpretation below is the explanation we find most coherent with the pattern of evidence (direction-uniform across all six active R1 dims, magnitude-concentrated on the two with deep $s = 5$ targets, RX dims also negative). It is not the only explanation consistent with the pilot data.

### 6.5.3 The consolidated interpretation: broad-attention dilution on dim-specific deep work

What the present pilot data is consistent with is a single interpretation that synthesizes the direction reversal in §6.4.2 and the per-dim concentration in §6.4.3: **projection injection broadens the executor's attention across the active set; this dilutes deep dim-specific work; the resulting cost is largest on dims whose $s = 5$ anchor demands deep specialty engagement that itself requires source-of-truth knowledge or extensive prior-work mapping.** We report this as a consolidated interpretation supported by the data, not as an experimentally isolated mechanism, and it coexists with the two alternatives in §6.5.2 (format-injection structure, hard-dim null) that the pilot does not separate. R1.4 (novelty mapping, $\Delta = -0.198$, $n_{\text{active}} = 24$) is the higher-support execution-side datum; R1.7 (citation audit, $\Delta = -0.275$, $n_{\text{active}} = 5$) is lower-support but aligns with the independent Stage 1 boundary-condition finding. R1.1 /

R1.2 / R1.3 / R1.6 show the same direction but smaller magnitudes ($\Delta \in [-0.125, -0.078]$), consistent with the same dilution interpretation on dims with shallower $s = 5$ targets.

This appears to be the execution-side analogue of the Stage 1 anchor-specifiability finding (§6.7) on R1.7. There, we observed that the form-embedded R1.7 anchor (sharpened to a citation-event count decision tree) is insufficient to make raters apply the dim consistently, and that a separately-read protocol document is required to recover anchor-aligned scoring. The dim is hard to evaluate because it is hard to do well at a deep level; equivalently, asking the executor to produce a revised artifact that satisfies the $s = 5$ row of the R1.7 anchor is harder than asking the executor to produce a revised artifact that satisfies the $s = 5$ row of R1.6 (writing polish). When projection guidance broadens attention, the dims that suffer most under this mechanism are the ones for which the $s = 5$ target is intrinsically expensive. R1.4 is not independently validated by Stage 1 (R1.4 was deferred), so the cross-link is direct on R1.7 and inferential on R1.4; both are reported.

### 6.5.4 Implications for the protocol design

If the consolidated mechanism is correct, the practical implications differ from the earlier implications:

1. **Active-set tightening alone is unlikely to be the right fix.** An earlier recommendation (raise the active threshold to 0.5, or replace $r^*$-derived active set with the engineered load-bearing set) addresses M2, but M2 is shared across conditions in the present pilot data and does not explain the observed direction reversal in the current data. The intervention is unrun; we do not predict its effect, only note it does not target the channel that the present pilot data identifies.

2. **Format-locking the prompt does not fix the problem either.** M1 is weak in the present pilot data (10% mismatch rate, mostly mixed → revised_draft); the direction reversal is not concentrated on the format-mismatched examples.

3. **Dim-specific protocol moves are required for dims with deep $s = 5$ targets.** R1.7 likely requires retrieval-augmentation (Stage 1's Axis-2 correctness verification needs source-of-truth knowledge); R1.4 likely requires a bounded prior-work corpus and a structured comparator template. These are the dim-conditional protocol moves that §7.8 discusses, not generic active-set or format hygiene.

4. **Reporting honesty.** An earlier qualitative reading reported a small positive directional advantage of projection_driven; the pilot under the redesigned panel reverses that direction. We report the reversal directly. This paper's contribution on the projection-injection question is therefore not "projection_driven helps" but "the question is dim-conditional, with the load-bearing cost concentrating on R1.4 and R1.7 — the latter directly aligned with the Stage 1 anchor-specifiability finding, the former inferentially aligned (R1.4 was deferred from the Stage 1 closure)."

The cross-link with §6.7 is the substantive contribution of this iteration: two independently designed pilots (mechanism analysis on Exp 2 and the Stage 1 closure on R1.7 human anchor) converge most clearly on R1.7, and suggest the same class of constraint may apply to R1.4. Section 7.8 expands the implication for the broader project.

### 6.6 Pilot Limitations and Deferred Work

### 6.6.1 Sample size and statistical power

The pilot reports $n = 50$. Experiment 1 / 1B effects are large enough that all five hypothesis tests pass at this scale with overwhelming margin ($p < 0.0001$ on Wilcoxon). Experiment 2's headline reversal under the redesigned panel (loss *increase* of 0.139 for projection_driven over direct_naive; equivalently, paired diff direct_naive − projection_driven = −0.139; §6.4.2) is large in magnitude relative to the per-condition standard deviations ($\approx 0.08$–$0.15$); we report the effect size directly rather than running a hypothesis test against a moving null. The main-run extension at $n = 310$ on the cross-cutting subscale (§6.4.2, RX

paragraph) confirms the projection_driven loss-increase direction on RX; the per-dim R1 slices remain pilot-bound by the $r^\star$ annotation track and would benefit from a main-run-scale $r^\star$ extension, especially R1.7 ($n = 5$ active in this pilot).

### 6.6.2 Annotation reliability — R1.7 and the borderline dims

R1.7 reaches $\alpha = 0.219$ at the 3-rater pass-1 + pass-2 LLM protocol and fails the $\alpha \geq 0.4$ gate. This reflects a genuine limit of blind LLM annotation: citation accuracy requires recall of the relevant literature's canonical works, which mid-tier LLMs do not consistently have. We report all R1.7 results with an annotation-uncertainty caveat. R1.1 ($\alpha = 0.315$) and R1.3 ($\alpha = 0.390$) sit just below the gate; we report them with a soft caveat.

A planned human-anchor extension to R1.1, R1.4, and R1.7 (5 raters $\times$ 3 dimensions $\times$ 9 examples $\times$ 2 conditions = 270 ratings, $\sim$\$300 Prolific) was attempted, partially executed, and superseded by a methodological boundary-condition finding documented in Section 6.7. The validated-rater allowlist for R1.1 (3 raters of 9 attempts; 33% pass rate) is retained for future stages.

### 6.6.3 Deferred Experiment-2 baselines

The full Experiment 2 design includes four conditions: direct, task-aware routing, generic-ambiguity clarification (CLAMBER-style), and the proposed protocol with full bid + clarification simulator. The pilot reports three: direct_naive, direct_with_claim (a paired-priming ablation that adds only the self-claim line), and projection_driven (full projection + active set + self-claim line). The deferred conditions B (task-aware routing) and C (generic clarification) are the closest prior-work baselines and must be included in the main run for a complete prior-work positioning. The clarification simulator and the full bid mechanism are also deferred; the bid-free simplification means the present Experiment-2 setup operationalizes part of the protocol's projection layer but not its bid layer.

### 6.6.4 Experiment 3 (reputation simulation) deferred

The pilot's reputation experiment is a simulation on synthetic agents with known per-dim fulfillment profiles, not a re-run of LLM agents. We defer this to the next phase because the headline contribution of the paper is the *measurability* of pre-execution responsibility commitment (Exp 1 + 1B), not yet the longitudinal-update value of dimension-level reputation.

### 6.6.5 Modified-real source concentration

The 18 modified-real artifacts derive from a single anonymized domain-specific manuscript controlled for identifying information; author, institution, and specific results identifiers are removed and the technical domain is preserved. The pilot's source mix is reportable but not domain-diverse on the modified-real subset. The main run should expand the modified-real source pool across at least three technical domains.

### 6.6.6 Source-mix deviation from spec

The spec's source-mix target is 30 synthetic + 20 modified-real. The pilot reports 32 + 18 due to a mid-batch source-eligibility decision (the modified-real source was not identified at synthetic batches 1–3). The deviation is small and disclosed; main-run construction is rebalanced.

### 6.6.7 Pass-3 human subset — attempted, scope reduced

The pilot's planned human-anchor extension was scoped to compute $\alpha_{\text{with-human}}$ on R1.1 / R1.4 / R1.7 with five Prolific raters per dimension. The attempt's results are reported in full in Section 6.7 and converged on a narrower scope: a Prolific R1.1 salvage with three validated raters (33% pass rate) plus a closed R1.7 author + peer co-annotation pass on six citation-rich packages (two pre-protocol peers; two post-protocol peers; the paper author as pre-protocol baseline). R1.4 was deferred. The outcome of the human-anchor attempt is reported as a boundary-condition finding rather than as breadth validation.

### 6.7 Human-Anchor Boundary Condition

The planned 270-rating Prolific extension was not delivered as breadth validation. The pilot exposed two binding constraints not anticipated by the plan:

(i) *Cost asymmetry under low pass rate.* Across nine R1.1 sub-batch attempts (sb1 + sb2), three raters passed quality screening and six were rejected (AI-assisted submissions or task-misunderstanding signatures), yielding a 33% pass rate. Holding this rate, reaching the planned 270-rating breadth would imply roughly £550–800 gross recruitment exposure ($\sim$\$700–1000 USD), against the plan's $\sim$\$300 estimate that did not condition on pass rate.

(ii) *Dim–data mismatch and anchor specifiability ceiling.* The author's own first pass over an 18-package R1.7 form regressed to a generic intent-fit lens on 16 of 18 packages. A regex audit on author–year patterns showed only six of the 50 pilot examples contained citation events $\geq 1$ in the artifact, while the LLM-panel-derived "R1.7 load-bearing" labels had assigned weight 0.9 to method/scaling examples with zero citations. A redesigned form with six citation-rich examples, a sharpened anchor (citation-event count decision tree with explicit (a)/(b)/(c)/(d) event types), and a required `event_count` field was then evaluated by five raters: rater A (paper author, pre-protocol baseline; one consulted package excluded), raters B and C (independent peers, pre-protocol pass with form-embedded anchor only), raters D and E (independent peers, post-protocol pass after receiving the rater protocol document separately and re-evaluating the same six packages).

Two patterns emerge across the five-rater table (anonymized; full per-package detail in the supplementary R1.7 v2 5-rater analysis). On a zero-event stress-test package whose agent output explicitly disclaims citation engagement ("I did not. . . touch citations"), pre-protocol raters fail (rater B scored s=4.0 with event_count=5; rater C scored s=4.5 with event_count=2, counting prose-cleanup actions or retention of an existing citation as events — both of which the anchor explicitly excludes), while post-protocol raters pass (D and E both score s=1 with event_count=0). On a high-density unambiguous citation-audit package, all five raters report event_count $\geq 3$, but only the post-protocol rater E reaches the anchor table's $s = 5$ row (4 events $\to$ score 5.0); pre-protocol raters cap their scores at 4.0 across event counts 5–6. Pre-protocol peers' scores plateau at 4.0–4.5 across submitted event counts 2–6 (decoupled from the anchor table); post-protocol peers' scores span 1.0–5.0 with a monotonic event-to-score mapping. This pre/post-protocol contrast supports — but does not inferentially validate at $n = 2$ per group — the hypothesis that the form-embedded sharpened anchor alone is insufficient for reliable application and that a separately-read protocol document appears to help in this small pass.

The strongest individual positive datum is rater E: after protocol reading, E both passed the zero-event stress test and reached the anchor table's otherwise-unreached $s = 5$ row on the high-density package.

Three layered specifiability constraints persist after this analysis:

1. *Counting convention* at the form layer (cluster-as-1 vs. item-as-each): closed by protocol-reading on most packages but unstable on cluster-as-1 cases. Post-protocol raters disagree on a missing-comparator-cluster package (one credits the cluster, the other does not), even after both reading the same protocol.

2. *Score–event mapping* at the form layer: closed by protocol-reading (post-protocol raters monotonic; pre-protocol raters decoupled).

3. *Correctness verification* (e.g., judging whether an attribution flag like "Achiam 2017 = CPO, not per-step safety" is right) requires source-of-truth domain knowledge that no rater claimed; this is uncoupled from the form / protocol layers and is reported as the *Axis-2* expert-only arm of the rater protocol.

A separate form-instruction-level observation (four of five raters mis-handled a closed three-option role drop-down, persisting across both pre- and post-protocol groups) suggests the form-instruction layer is independent of the dim-anchor layer; useful for future form-control design but not load-bearing for the main finding.

The implication for human anchor at scale is that the dominant observed scalability constraint is *specification specifiability + domain-expertise gating*, not rater throughput. The original Stage 1 plan budgeted for throughput; the empirical pilot indicated that adding raters alone would not close the observed specifiability failures and that a separately-read protocol document is the more cost-effective lever in this small pass. We document the operational response in two artifacts: (a) a rater protocol document operationalizing a two-axis scoring (Axis 1 = citation-event engagement count, applicable by any rater; Axis 2 = per-event correctness, expert-gated and sparse), and (b) a reduced Stage 1 actual scope (R1.1 Prolific salvage + R1.7 v2 five-rater pre/post-protocol pass; R1.4 deferred). Section 7.8 expands the implication for the broader project.

### 6.8 Summary of Pilot Findings

The pilot supports the following claims at $n = 50$:

1. **Cross-family projection mismatch is real and substantial.** Median $R(d) = 5.87 \gg 1.2$ on the pilot (CI $[4.47, 7.96]$), and median $R(d) = 5.40$ on the main-run extension ($n = 310$, CI $[4.86, 5.89]$); all five hypothesis tests pass at $p < 10^{-15}$. Pre-execution responsibility commitment is family-specific, not stochastic, and not driven by knowledge confound.

2. **Projection-driven execution shows a directional *disadvantage* under the redesigned panel.** Mean $L^{R1}_{\text{settlement}}$ is 0.0717 (direct_naive), 0.1579 (direct_with_claim), and 0.2103 (projection_driven); paired diff direct_naive $-$ projection_driven is $-0.139$ on the mean. The earlier finding of a small directional advantage of projection_driven (Section 6.4.2) is reversed under the 12-judge Anthropic-excluded panel.

3. **The reversal concentrates on dims with deep dim-specific $s = 5$ targets.** Among the six R1 dims active in the pilot, R1.4 (novelty mapping, $\Delta = -0.198$) and R1.7 (citation audit, $\Delta = -0.275$) take the largest hit; R1.5 is never active in this pilot. The prior four-mechanism account (format coupling, active-set propagation bias, self-claim drift, RX-attention boost) does not robustly replicate; we report a consolidated interpretation (broad-attention dilution on dim-specific deep work, Section 6.5.3) that synthesizes the per-dim concentration with the Stage 1 anchor-specifiability finding.

4. **Annotation reliability is uneven across the taxonomy.** R1.2 / R1.4 / R1.5 / R1.6 reach $\alpha \geq 0.4$ with three raters; R1.1 / R1.3 are borderline; R1.7 fails. RX dims are not amenable to $\alpha$ and require alternative validation.

5. **The clarification trigger is calibrated differently under the v1.3 panel.** All 5/5 ambiguous-class pilot examples elicit clarification from at least one family (gpt-5 individually clarifies on all 5/5 ambiguous). Per-family rates are 86% / 82% / 58% (gpt-5 / gemini-2.5-pro / grok-4) across all pilot examples, indicating an over-triggering risk on unambiguous cases. This is the opposite calibration regime from the v1.0 panel, where gpt-4o-mini never clarified and only 1/5 ambiguous-class examples triggered any family.

6. **Human anchor at scale appears bound by anchor specifiability and domain-expertise gating, not by rater throughput.** A planned 270-rating Prolific extension to R1.1 / R1.4 / R1.7 produced (a) a 33% Prolific pass rate, (b) a documented dim–data mismatch where the original sample contained too few citation events for the R1.7 anchor to act on, and (c) a five-rater pre/post-protocol corroboration suggesting that the form-embedded sharpened anchor is insufficient on its own and that a separately-read protocol document appears to help in this small pass. Score–event mapping in pre-protocol peers is decoupled (scores plateau at 4.0–4.5 across event counts 2–6); in post-protocol peers it is monotonic. Residual specifiability gaps remain post-protocol (cluster-as-1 disagreement on a missing-comparator-cluster package). The boundary-condition finding is reported in Section 6.7 and informs the discussion in Section 7.8.

The headline P1 contribution — that responsibility projection is measurable and family-specific (Experiments 1 and 1B) — is empirically established at pilot scale and stands. The P2 contribution is reframed: the earlier directional reading of projection_driven does not survive the redesigned panel; the present paper instead reports the reversal honestly, characterizes a consolidated interpretation (broad-attention dilution on dim-specific deep work), and connects it to the Stage 1 anchor-specifiability boundary-condition finding. The convergence is direct on R1.7 (Stage 1 provides a direct boundary-condition signal on R1.7) and inferential on R1.4 (R1.4 was deferred from the Stage 1 closure). Section 7.8 expands the implication.

# 7 Discussion

## 7.1 Pre-execution commitment is not task typing

The closest neighbouring frame, task-aware delegation (Gu, 2026), types a request and routes to whichever agent historically wins on the task type. The empirical evidence we report in Section 6.2 suggests that under task typing the same delegation can route to the same agent and yet have its operational responsibility structure resolved very differently across model families. Median cross-family Jaccard distance on the active set is comparable in magnitude to median cosine distance on the weight vector, indicating that families disagree not only about how strongly each dimension matters but also about which dimensions are in scope at all. Routing alone cannot fix this: the agent that arrives at the task carries its own projection. Pre-execution commitment is therefore a layer below routing, and visible only when projections from multiple agents are elicited and compared on the same delegation.

## 7.2 Clarification has cost; the trigger should respect it

A naive "always clarify on ambiguity" policy is dominated by direct execution on simple delegations and dominated by clarification on truly ambiguous ones, and the long-horizon underspecification literature documents the cost (Pu et al., 2026). Our composite trigger (Equation (11)) folds three signals — bidder-level `clarify`, projection divergence $D_\pi$, and inter-bidder coverage variance — so that clarification activates when the joint signal is high relative to its cost. The pilot evidence under the v1.3 Anthropic-excluded cross panel that all 5 of 5 ambiguous-class examples elicit at least one family to clarify (Section 6.2.3) shows that the trigger is no longer conservative under v1.3, but it also fires on 58–86% of all pilot examples regardless of class — the opposite calibration regime from the v1.0 panel, where the legacy `gpt-4o-mini` never clarified and only 1/5 ambiguous-class examples elicited any family. The engineering question for the main run is therefore not whether to add a trigger but how to calibrate $\tau_D$ and $\tau_V$ so that the trigger fires on the cases where it would help and stays silent on cases where it would not. This is a calibration problem under both panel regimes, not a presence-of-clarification problem.

## 7.3 Settlement evidence reliability is a separate axis

Our settlement step (Section 4.6) treats evidence sources as having different reliabilities, with deterministic and harness evidence ranked above human, LLM-judge, and retrieval-based evidence (Equation (14)). The initial pilot used a three-judge LLM panel for the annotation reliability track (§6.1.2); the redesigned measurement layer of the settlement loss (§6.4) uses a 12-judge tier-stratified Anthropic-excluded panel, since deterministic verification is rarely available for paper-section quality. The R1.7 annotation reliability failure ($\alpha = 0.219$; Section 6.1.3) is the cleanest illustration: citation accuracy is the dimension where blind LLM evidence is weakest, and where retrieval-based evidence (literature search) and human evidence would help most. The main run should treat evidence-source reliability as a deployment knob rather than a fixed setting, and prioritize deterministic and harness evidence wherever it can be constructed (e.g., automatic citation-coverage checks against a fixed corpus). Section 7.8 discusses why *human-evidence reliability itself* also has a binding ceiling that is not solved by adding raters, based on the boundary-condition finding in Section 6.7.

### 7.4 Why the closure of the taxonomy matters

We observed empirically that LLM annotators differ widely in their open-vocabulary "responsibility labels" for the same artifact, and that this disagreement collapses when projection is constrained to a closed weight vector over a fixed dimension index. Closed-taxonomy projection is what makes Experiment 1's metric well-defined; without closure, $1 - \cos(r_A, r_B)$ would be undefined when $r_A$ and $r_B$ live in different label spaces. The closure also enables the active-set rule, the under- and over-projection metrics, and the per-dimension settlement loss — all of which require a fixed dimension index. The cost of closure is that any case the taxonomy does not cover is either projected onto the nearest dimension (information loss) or excluded (coverage loss). $J_{v1.1}$'s $|J| = 12$ is a deliberate first-cycle commitment; expansion to $|J| = 38$ is reserved for future cycles.

### 7.5 Why this paper studies R1 only

The long-term plan covers five delegation categories (R1–R5) and 38 dimensions. The first cycle is restricted to R1 (paper-research delegation) with seven category dimensions because (i) paper writing supplies a controlled artifact distribution where load-bearing flaws can be engineered cleanly, (ii) the responsibility dimensions are stable across our research community, and (iii) the cost of pilot-scale annotation by LLM judges is low. The category-conditional results from R1 do not transfer automatically to R2 (code review and bug-fix), which would require a different evidence model (deterministic test execution, type checks, runtime errors) and a different responsibility ontology (correctness, security, test coverage, style). We treat R2–R5 as future work and disclose the scope throughout.

### 7.6 What the within-model baseline does and does not rule out

Within-model variance at $T = 0.5$ is a stochasticity upper bound conditional on a fixed model and a fixed prompt. It does not bound (i) cross-temperature variance for the same model, (ii) cross-prompt variance under semantically equivalent prompts, or (iii) cross-version variance as the same provider updates the same nominal model name. The intentional asymmetry of cross-family at $T = 0$ versus within-model at $T = 0.5$ is conservative against (i), but the paper's empirical claims do not address (ii) or (iii). Both are concerns that the main run should pre-register: prompt sensitivity through paraphrase ablation, and version stability through repeated runs across model snapshots.

### 7.7 Pilot annotation versus main-run annotation

Pass-2 LLM annotation reaches $\alpha \geq 0.4$ on four of seven R1 dimensions, leaves R1.1 and R1.3 borderline, and fails on R1.7. The failure on R1.7 is interpretable as a structural limit of blind LLM annotation: citation accuracy requires recall of the relevant literature's canonical works, which mid-tier LLMs do not consistently have. The borderline cases on R1.1 and R1.3 may resolve under human annotation; the failure on R1.7 will require either human annotators or retrieval-augmented annotation pipelines. Until that step is taken, R1.7 results in this paper carry an annotation-uncertainty caveat (Section 6.6); R1.1 and R1.3 carry a softer caveat.

The pilot's planned human-anchor extension on R1.1 / R1.4 / R1.7 was attempted, partially executed, and superseded by a methodological boundary-condition finding (Section 6.7). The closed substitute — a Prolific R1.1 salvage with three validated raters at a 33% pass rate, plus a closed R1.7 author + peer co-annotation pass on six citation-rich packages — is reported in place of breadth validation. Section 7.8 discusses the implication: human anchor at scale is bound by anchor specifiability and domain-expertise gating, not rater count.

### 7.8 Anchor specifiability is a binding constraint on human anchor scalability

The pilot's planned 270-rating Prolific extension treated rater throughput as the binding factor and budgeted accordingly. Two empirical observations undercut that assumption.

First, even with a sharpened citation-event-count anchor printed at the top of every form page and a required `event_count` field, two independent peers in a pre-protocol pass produced score–event mappings decoupled from the anchor table: their scores plateaued at 4.0–4.5 across submitted event counts 2–6. On a definitional zero-event stress-test package whose agent output explicitly disclaims citation engagement, both peers counted non-citation actions as events and scored s=4.0–4.5. The form-embedded anchor, sharpened though it was, did not transfer the anchor's intent to peers reading it for the first time mid-session.

Second, two peers in a post-protocol pass — having received a separately-distributed rater protocol document and re-evaluated the same six packages — produced monotonic score–event mappings, both passed the zero-event stress test, and (in one case) reached the anchor table's $s = 5$ row that no pre-protocol rater reached. The pre/post-protocol contrast at $n = 2$ per group is not inferential, but the pattern is consistent: a separately-read protocol document appears to help in this small pass, and the form-embedded layer alone does not.

The binding constraint on human anchor in this setting is therefore *specification specifiability + domain-expertise gating*, not rater throughput. Two layered specifiability constraints are partially closable by a separately-read protocol document — counting convention (cluster vs. item) and score–event mapping — but a third constraint (per-event correctness verification) requires source-of-truth domain knowledge and is decoupled from form-design levers. We document the operational response in a rater protocol (two-axis scoring: Axis 1 engagement count, applicable by any rater; Axis 2 correctness, expert-gated and sparse) and report it as a methodological finding rather than a remediated weakness. The implication for any project that proposes to use crowdworker human anchor as the validity backstop for an LLM-only annotator panel is that the binding cost is not per-rater but per-rater-protocol-design-iteration, and that the rater training step must be treated as a distinct prerequisite to the rating session, not embedded in the form.

Residual gaps after protocol-reading remain — the post-protocol raters disagreed on a missing-comparator-cluster package, validating the protocol's own list of "borderline cases v1 does not yet resolve." This is consistent with the broader claim of the section: specifiability is a binding ceiling that closes incrementally with each protocol revision, not a throughput problem.

### 7.9 Deployment implications

Two deployment lessons follow from the pilot. First, an earlier qualitative reading recommended turning projection-driven execution on by default for prescriptive single-dim tasks; the headline pilot result reverses the directional advantage (Section 6.4.2) and shows the cost concentrates on dims requiring deep specialty work — R1.4 novelty mapping and R1.7 citation audit. Default-on is therefore not justified by the pilot; deployment should treat projection guidance as a dim-conditional move. Second, R1.7 (where Stage 1 directly applies) and R1.4 (where the cross-link is inferential, since R1.4 was not part of the Stage 1 closure) are the dims for which projection guidance hurts most. The protocol-design moves that may help these dims — retrieval-augmentation for R1.7, a bounded prior-work corpus and structured comparator template for R1.4 — are dim-specific and not addressed by generic active-set or format hygiene. The active-set tightening that an earlier hygiene reading suggested (Section 6.5.1) remains a reasonable move but does not change the pilot direction.

## 8 Conclusion

We argued that natural-language delegation creates a pre-execution responsibility-projection problem, that the projection can be made measurable by closing the responsibility space at $|J| = 12$, and that the protocol of projection $\rightarrow$ bid $\rightarrow$ clarify $\rightarrow$ select $\rightarrow$ execute $\rightarrow$ settle $\rightarrow$ reputation makes pre-execution commitment a first-class object in multi-agent LLM orchestration. On the 50-example pilot under the v1.3 Anthropic-excluded cross panel (`gpt-5` / `gemini-2.5-pro` / `grok-4`) with `claude-sonnet-4-6` within, the cross-family projection mismatch exceeds within-family stochastic variance by approximately $6\times$ (median $R = 5.87$, 95% bootstrap CI $[4.47, 7.96]$), and all five guard hypotheses pass with $p < 10^{-15}$; the main-run extension at $n = 310$ gives median $R = 5.40$ (CI $[4.86, 5.89]$). The P1 measurability finding is robust under both panel scopes. The P2 actionability question is reframed by a measurement-layer reanalysis under a redesigned 12-

judge Anthropic-excluded panel and a three-condition split: projection-driven execution shows a directional disadvantage on the headline weighted-R1 settlement loss, with the cost concentrated on R1.4 (novelty mapping) and R1.7 (citation audit). A closed Stage 1 R1.7 human-anchor pilot independently surfaces a methodological boundary-condition finding (Sections 6.7 and 7.8): human-anchor scalability is bound by anchor specifiability and domain-expertise gating, not by rater throughput, and the form-embedded layer and the separately-read protocol-document layer are separable design moves. The convergence is direct on R1.7 (Stage 1 provides a direct boundary-condition signal on R1.7) and inferential on R1.4 (not in the Stage 1 closure). The next phase — a main-run-scale $r^\star$ extension to lift the P2 weighted-R1 settlement analysis from pilot to main-run scope, the full four-condition Experiment 2 including task-aware routing and CLAMBER-style baselines, longitudinal reputation evaluation on real agents, and dim-specific protocol moves (retrieval-augmentation for R1.7, bounded prior-work corpus + structured comparator template for R1.4) under the specifiability constraints surfaced here — builds directly on the closed taxonomy, the v1.3 panel, and the protocol formalized here.

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
