# OpenReview forum: "When Responsibility Guidance Hurts: A Pilot Study of PreExecution Projection in LLM Agents"
_TMLR — Under review for TMLR_

### Review · Reviewer_DVew · 2026-06-04

**Summary Of Contributions:**

This paper studies what the authors call pre-execution responsibility projection in LLM agents: before an agent acts on an underspecified delegation, such as “improve this paper draft,” it implicitly decides which responsibilities are in scope, such as conceptual reconstruction, novelty assessment, writing polish, or citation audit. The paper formalizes this projection as a multi-label weight vector over a closed 12-dimensional taxonomy for paper-research delegation. It then proposes a broader protocol: project responsibilities, solicit responsibility-bearing bids, trigger clarification when needed, select an agent, execute, settle performance per dimension, and update per-dimension reputation.

**Audience:**

Yes

**Audience Explanation:**

The paper addresses a timely issue for LLM agents: underspecified delegation and the gap between what a user asks, what an agent assumes it has been asked to do, and what the agent eventually executes. This is relevant to researchers working on agent orchestration, clarification, routing, LLM-as-judge evaluation, benchmark design, and human-AI collaboration.

**Broader Impact Concerns:**

The main ethical concern is that responsibility projection could create a false sense of accountability: users may believe an agent has reliably understood and committed to the intended responsibilities when the projection is actually model-specific, weakly validated, or wrong. This is especially risky for high-stakes domains such as research evaluation, citation auditing, peer review, legal writing, medical summarization, or safety-critical code review.

**Claims And Evidence:**

No

**Claims Explanation:**

1. The paper proposes a full responsibility-bearing delegation protocol, but the empirical evaluation only tests the projection layer and a simplified projection-injection treatment. The bid mechanism, clarification simulator, task-aware routing baseline, generic clarification baseline, and longitudinal reputation update are all deferred. This means the paper does not yet show that the proposed protocol improves orchestration, reliability, or accountability in deployed multi-agent systems.
2. The paper uses a median of one author annotation and two LLM annotators, but annotation reliability is uneven. R1.7 fails reliability with α=0.219, R1.1 and R1.3 are borderline, and the median-derived active set substantially disagrees with the engineered labels. In particular, R1.5 is engineered in 8 examples but never active under the median-derived labels, while R1.2, R1.3, and R1.4 are over-activated relative to the engineered labels. Because the execution analysis depends on this active-set and weighting scheme, these annotation issues materially affect the
3. The P2 actionability result is interesting but should be treated as a pilot finding rather than a strong conclusion about responsibility guidance. The execution experiment uses only 50 examples and three conditions, with a single executor family and a 12-model LLM-judge panel. The authors themselves acknowledge a possible reverse-bias issue because the executor is Anthropic-family while the judges are non-Anthropic. The result is still valuable as a negative pilot finding, but it is not enough to establish a general boundary condition for projection-driven execution.

**Requested Changes:**

1. The paper should more clearly distinguish between what is empirically shown and what is only proposed. Please check my answer for "Are the claims made in the submission supported by accurate, convincing and clear evidence?"
2. The closed taxonomy is central to the paper, but it appears mostly author-designed. The authors should provide evidence that the 12 dimensions are understandable

---

> ### Author Response · Authors · 2026-07-13
>
> [Response to Reviewer DVew — Part 1/4]
>
> We thank Reviewer DVew for the careful reading and the specific quantitative concerns. Before the point-by-point response, we must disclose a reporting error we found while auditing the reliability statistics the reviewer questioned.
>
> Correction of a reporting error in the submitted reliability table
>
> During our audit, we discovered that the submitted reliability table was generated from a superseded annotator track. This was our reporting error, and we apologize. Although the reported P1/P2 numerical outputs were computed from the intended artifacts, the correction materially weakens the reliability evidence supporting the weighted-R1 interpretation.
>
> - What happened. Early in the pilot, r* annotation used one author plus two closed-API LLM annotators (claude-sonnet-4.5 and gemini-2.5-flash). That track was superseded mid-pilot by the canonical track: one author plus three open-weight annotators (llama-3.1-70b, qwen-2.5-72b, mistral-large-2). The submitted Table 2 (tab:alpha) was generated from the superseded track's statistics file rather than from the canonical file, and the submission's description of the annotation setup ("one author and two LLM annotators") likewise describes the superseded roster. The reviewer's reading of the setup therefore reflects our error, not a misreading.
> - Corrected values. Under the canonical track, reliability is weaker than submitted: only R1.6 passes the α ≥ 0.4 gate (0.585); R1.1 = 0.105, R1.2 = −0.011, R1.3 = 0.080, R1.4 = 0.202, R1.5 = −0.044, R1.7 = 0.031. The submitted characterization "R1.7 fails; R1.1 and R1.3 borderline" is superseded: six of seven R1 dims fail the gate.
> - What each reported result derives from. P1's separation ratio R = d_C/d_W is computed from projection vectors and uses neither r* nor α. P2's weighted-R1 loss uses the canonical r*-median file (author + 3 open-weight annotators), which was always the file used for the reported analysis. No headline number is recomputed by this correction — but the reliability evidence standing behind the weighted-R1 quantities is weaker than the submission implied, and every non-R1.6 R1 result now carries an explicit annotation-uncertainty caveat.
> - Revision and notification. The revision corrects the table and all dependent sentences and interpretations, and adds the full annotator-panel chronology. We are also notifying the Action Editor of this correction separately.
>
> Point 1 — Full protocol vs projection layer scope
>
> Reviewer:
> > The paper proposes a full responsibility-bearing delegation protocol, but the empirical evaluation only tests the projection layer and a simplified projection-injection treatment. The bid mechanism, clarification simulator, task-aware routing baseline, generic clarification baseline, and longitudinal reputation update are all deferred.
>
> Response: The reviewer is correct, and the submission's contribution positioning could imply broader evaluation than we performed. The revision adds a scope clause to the protocol section with the following table:
>
> Protocol evaluation status
>
> | Step | Status | Notes |
> |---|---|---|
> | 1. Projection | pilot evaluated | Exp 1, n=50 + n=310 |
> | 2. Bid | formalized only, not evaluated end-to-end | |
> | 3. Clarify | descriptively instantiated | trigger rate reported; simulator and calibrated cost not run |
> | 4. Select | formalized only, not evaluated end-to-end | |
> | 5. Execute | pilot evaluated | Exp 2, 3 conditions |
> | 6. Settle | settlement metric operationalized in the pilot | per-dim v_ij, weighted-R1 loss |
> | 7. Reputation | formalized only, not evaluated end-to-end | |
>
> We do not provide end-to-end evidence that the seven-step protocol improves orchestration, reliability, or accountability. The revision states this sentence explicitly, rewords the closed-responsibility-taxonomy contribution bullet to "we formalize a protocol; this paper pilot-evaluates Steps 1 and 5, operationalizes the settlement metric used for Step 6, and descriptively instantiates Step 3," and moves steps 2, 4, and 7 — together with the bid mechanism, clarification simulator, task-aware routing, and a CLAMBER-style baseline — to named future-work items.
>
> (continued in the next comment)

---

> ### Author Response · Authors · 2026-07-13
>
> [Response to Reviewer DVew — Part 2/4]
>
> Point 2 — Annotation reliability + engineered vs median active set mismatch
>
> Reviewer:
> > R1.7 fails reliability with α=0.219, R1.1 and R1.3 are borderline, and the median-derived active set substantially disagrees with the engineered labels. In particular, R1.5 is engineered in 8 examples but never active under the median-derived labels, while R1.2, R1.3, and R1.4 are over-activated relative to the engineered labels. […]
>
> Response: The reviewer's numerical observation is accurate — R1.5 is engineered in exactly 8 pilot examples and 0 are active under median r* > 0.3. The reliability half of this point is answered by the correction above: under the canonical track the concern is stronger than the reviewer stated, since six of seven R1 dims fail the gate.
>
> Engineered-set sensitivity analysis. To address the active-set-construction critique, we re-judged 1,800 tuples (50 examples × 3 conditions × 12 judges) with the engineered active set as prompt context, under two weighting schemes fixed in advance. Procedure: the executor outputs are unchanged (the same execution files are re-scored); what changes is the judge prompt's active-set context, so this is a re-judging sensitivity, not a reweighting of existing scores; the bootstrap resamples examples.
>
> | weighting | paired diff (projection_driven − direct_naive) | 95% bootstrap CI |
> |---|---|---|
> | headline (median active R1) | +0.139 | [+0.109, +0.170] |
> | W-uniform (engineered R1 only) | +0.126 | [+0.086, +0.175] |
> | W-median (r* restricted to engineered R1) | +0.107 | [+0.069, +0.147] |
>
> Both schemes preserve the direction (positive sign, CI excludes 0); the magnitude shrinks by 9–23%, so we do not claim independence from the active-set construction. (W-median retains n = 45/50 paired examples — five examples carry zero r* weight on all engineered R1 dims; W-uniform retains n = 50.)
>
> Panel-sensitivity diagnostic. An alternative three-family re-annotation of all 50 pilot delegations, run under a protocol and decision rule fixed before any annotation call, is reported in the revision's annotation-reliability subsection. Its outcome is consistent with panel sensitivity but does not identify the cause of the panel difference (annotator capability vs residual anchor ambiguity), and it does not repair the canonical reliability numbers: the canonical open-weight-panel table remains the paper's primary reliability evidence, and the per-dim caveats stand:
>
> | dim | active in median r* set | reliability (canonical) |
> |---|---|---|
> | R1.1 | 15/50 (~30%) | fails α (0.105) |
> | R1.3 | 20/50 (~40%) | fails α (0.080) |
> | R1.7 | 5/50 (~10%) | fails α (0.031) |
>
> R1.6-only sensitivity. Because only R1.6 passes the corrected reliability gate, we also restricted the PD − DN comparison to R1.6 alone (active in 16/50 examples): paired v_ij difference −0.078 (v_ij is the per-dimension fulfillment score, so the negative sign means lower fulfillment under PD), 95% bootstrap CI [−0.172, 0.000], Cohen's d −0.444; PD never outperforms DN on R1.6, degrades 3/16 examples, and the remaining 13 are tied at ceiling. Reported descriptively — directionally consistent with the headline loss, but marginal and ceiling-limited. Accordingly, the revision states the P2 headline's status explicitly: a descriptive pilot result measured against a low-reliability operational target.
>
> Revision commitment: corrected reliability table + panel chronology; the engineered-set sensitivity table in the Experiment 2 results; the panel-sensitivity diagnostic in the annotation-reliability subsection; the R1.6-only sensitivity and the P2 status sentence in the Experiment 2 results; per-dim reliability caveats and per-dim CIs in the per-dimension breakdown (see also our response to Reviewer E2Uz, Requested Change 3).
>
> (continued in the next comment)

---

> ### Author Response · Authors · 2026-07-13
>
> [Response to Reviewer DVew — Part 3/4]
>
> Point 3 — P2 pilot scale + executor family reverse-bias
>
> Reviewer:
> > The execution experiment uses only 50 examples and three conditions, with a single executor family and a 12-model LLM-judge panel. The authors themselves acknowledge a possible reverse-bias issue because the executor is Anthropic-family while the judges are non-Anthropic.
>
> Response: We agree the P2 result is pilot-scale and reserve broader evaluation for follow-up. The revision contains a limitation subsection ("family-level judge bias and single-executor design") stating this directly: the executor is claude-opus-4-7 (Anthropic family) while the 12-judge panel excludes Anthropic; this prevents self-scoring but does not remove family-level scoring bias, whose direction is not estimated in this pilot; and Experiment 2 uses a single executor family by design so that the three conditions are paired on identical executor capability.
>
> On the related "bundled interventions" concern, our ablation program decomposes the treatment — see our response to Reviewer zavj, Point 2, for the full breakdown, including the component matrix. All ablation conditions are built on top of the self-claim baseline (direct_with_claim, DWC), so each X − DWC row is an incremental contrast relative to DWC, conditional on the components shared with DWC — not an isolated component effect. A1 − DWC, A5 − DWC, and W1 − DWC have CIs including 0; NL1 − DWC and F1 − DWC exclude 0 but cross the panel boundary and remain exploratory. Separately, the DWC − DN treatment difference is +0.086 [+0.056, +0.116]; DWC adds only the self-claim JSON line to direct_naive. The results provide an exploratory pattern consistent with a joint form/content interaction, but they do not identify the causal mechanism; a same-panel confirmatory run is committed future analysis.
>
> Revision commitment: the limitation subsection above (already in the revised manuscript); full ablation decomposition in the Experiment 2 results; multi-executor extension as future work.
>
> Point 4 — Requested Change: taxonomy interpretability evidence
>
> Reviewer:
> > The closed taxonomy is central to the paper, but it appears mostly author-designed. The authors should provide evidence that the 12 dimensions are understandable […]
>
> Response: The taxonomy is author-designed, and we do not claim 12-dimension human-interpretability evidence. What the data supports:
>
> - One dimension met the LLM-panel agreement threshold: only R1.6 passes α ≥ 0.4 under the canonical track (0.585). This is LLM-panel agreement, not human-interpretability evidence.
> - Not supported at this panel tier (6 R1 dims): R1.1–R1.5 and R1.7 fail the gate. The failure pattern — the most surface-observable dimension passing while dimensions requiring deeper interpretation show low agreement — is consistent with panel sensitivity, but annotator capability and genuine anchor ambiguity cannot be separated at this tier. R1.7 suggests a possible anchor-specifiability limitation (anchor-specifiability discussion).
> - RX dimensions (5 cross-cutting): carry less interpretability evidence, since RX scoring is operationalized as cross-cutting rather than example-specific.
>
> The revision adds a "Taxonomy interpretability evidence" subsection stating exactly this, presented as acknowledgment and re-scoping, not resolution: the concern stands, and a human interpretability study at adequate scale is committed follow-up work. We also cite Sadallah et al. (EMNLP 2025, RevUtil) as the closest closed-aspect-taxonomy methodological cousin, noting that RevUtil's α values are over human annotators and are not comparable to our LLM-annotator track.
>
> (continued in the next comment)

---

> ### Author Response · Authors · 2026-07-13
>
> [Response to Reviewer DVew — Part 4/4]
>
> Point 5 — Broader Impact: false accountability + high-stakes domains
>
> Reviewer:
> > Responsibility projection could create a false sense of accountability: users may believe an agent has reliably understood and committed to the intended responsibilities when the projection is actually model-specific, weakly validated, or wrong. Especially risky for high-stakes domains such as research evaluation, citation auditing, peer review, legal writing, medical summarization, or safety-critical code review.
>
> Response: The revised Broader Impact subsection now enumerates five named risks — model self-report is not external validation; overtrust (the Experiment 2 result shows that injecting responsibility information did not ensure improved fulfillment); high-stakes domain harm, listing the same domains the reviewer named; accountability laundering (an explicit projection can diffuse responsibility across model/developer/operator/user, and logs are not evidence of accountability satisfaction); and reputation amplification of judge-panel bias — plus seven deployment disclaimers (advisory not guarantee; per-dimension audit of recognition and fulfillment; independent verification and human oversight in high-stakes domains; per-model calibration evaluated separately; projection logs are not legal/ethical accountability evidence; deployed systems should provide the user with inspect/challenge/revise capability; and system-level phrasing "assumes responsibility, pending verification").
>
> Close
>
> To be explicit about status: the taxonomy-interpretability, annotation-reliability, and single-executor-generalization concerns are not resolved by the analyses above. They are acknowledged, quantified where the data permits, and the paper's claims are re-scoped accordingly — and the reporting-error correction leaves the reliability evidence weaker than the submission presented. Their resolution (a human interpretability study, a higher-reliability annotation track, a multi-executor extension) is committed follow-up work, not a discussion-phase deliverable. We welcome follow-up questions during the remainder of the discussion phase.

---

### Review · Reviewer_zayj · 2026-06-11

**Summary Of Contributions:**

This paper introduces the notion of pre-execution responsibility projection in LLM agents: before an agent acts on an underspecified natural-language delegation, it implicitly commits to a particular structure of responsibilities. The authors formalize this projection as a multi-label weight vector over a closed 12-dimensional taxonomy for paper-research delegation, and study whether different model families project different responsibility structures for the same delegation.

The main empirical contribution is a measurement result: cross-family responsibility projections differ substantially more than within-model stochastic variation. This result is supported both on the 50-example pilot and on a larger n=310 extension. The paper also reports an important negative result: naively injecting the projected responsibility vector and active set into the execution prompt hurts performance relative to direct execution, especially on dimensions such as novelty mapping and citation audit. A further methodological contribution is the human-anchor pilot, which suggests that reliable human annotation for some dimensions, especially citation audit, is limited by anchor specifiability and domain expertise rather than merely by rater throughput.

Key strengths include the originality of the pre-execution responsibility-projection framing, the use of a closed taxonomy that makes cross-model comparison possible, the cross-family vs. within-model baseline comparison, and the transparent reporting of negative and failed pilot results. Key weaknesses include the narrow task scope, pilot-scale evaluation for the actionability claims, imperfect ground-truth reliability, treatment confounds in the projection-driven execution experiment, and limited evidence for the human-anchor scalability claim.

**Additional Comments:**

No additional comments

**Audience:**

Yes

**Audience Explanation:**

The paper identifies a meaningful and underexplored layer in LLM-agent systems: the responsibility structure an agent implicitly adopts before execution. This is relevant to researchers working on LLM agents, multi-agent orchestration, task routing, clarification, LLM-as-judge evaluation, and accountability in delegated AI systems.

The finding that different model families project substantially different responsibility structures for the same delegation is likely to interest the TMLR audience, especially because it suggests a failure mode that is neither standard routing failure nor post-hoc output failure. The negative result is also useful: the paper shows that making responsibility projections explicit does not automatically improve execution and may hurt performance on dimensions requiring deeper domain-specific engagement.

**Broader Impact Concerns:**

The paper does not raise severe broader-impact concerns.

**Claims And Evidence:**

No

**Claims Explanation:**

The paper’s strongest claim, namely that responsibility projection is measurable and family-attributable, is reasonably well supported. The comparison between cross-family projection distance and within-model stochastic variation is clear, and the n=310 extension strengthens this claim. The reported median ratio R(d) is 5.87 on the pilot and 5.40 on the larger extension, with confidence intervals and paired tests supporting the direction of the effect.

However, I do not think all of the paper’s claims are supported with equally convincing evidence. The actionability claim is much weaker. The projection-driven execution experiment is only pilot-scale and combines several interventions at once: projected weights, active set, and a self-claim line. Although the paper includes a direct-with-claim condition, the design still does not fully isolate whether the observed degradation is caused by the projection content itself, the active-set presentation, prompt-format changes, or other priming effects. The authors also defer important baselines, including task-aware routing, generic clarification, the full bid mechanism, and CLAMBER-style clarification, to future work.

The reliability of the operational ground truth is another concern. The paper reports uneven annotation reliability: four R1 dimensions pass the stated alpha threshold, R1.1 and R1.3 are borderline, and R1.7 fails. Since R1.7 is also one of the dimensions emphasized in the actionability and human-anchor conclusions, this weakens the strength of the evidence.

Finally, the human-anchor claim is interesting but based on a very small and partially executed pilot. The pre/post-protocol comparison has only two raters per group, and the authors themselves note that it is not inferential. This is useful as a methodological warning, but not yet convincing evidence for a general “anchor specifiability ceiling.”

Overall, I find the P1 measurement claim substantially supported, but the broader actionability and human-anchor scalability claims remain pilot-level and should be stated more cautiously.

**Requested Changes:**

1.The paper should more clearly separate what is strongly supported from what is only pilot-level. The P1 measurability claim is well supported, but the P2 actionability and P3 human-anchor claims should be framed more cautiously. In particular, the paper should avoid implying that responsibility guidance generally hurts; the evidence only shows that the current projection-injection implementation hurts in this specific setting.
2. The current projection-driven condition bundles multiple changes: projection vector, active set, and self-claim line. The authors should add cleaner ablations or clearly acknowledge that the causal source of the degradation is not isolated. Useful ablations would include: active set only, weights only, self-claim only, natural-language responsibility summary only, and a format-controlled projection prompt.
3. The paper defers task-aware routing, generic clarification, the full bid mechanism, and CLAMBER-style baselines. If these cannot be included, the paper should more clearly state that it evaluates only the projection measurement layer and a limited projection-injection treatment, not the full proposed protocol.
4. The paper defers task-aware routing, generic clarification, the full bid mechanism, and CLAMBER-style baselines. If these cannot be included, the paper should more clearly state that it evaluates only the projection measurement layer and a limited projection-injection treatment, not the full proposed protocol.

---

> ### Author Response · Authors · 2026-07-13
>
> [Response to Reviewer zavj — Part 1/4]
>
> We thank Reviewer zavj for the structured reading and for separating the P1 measurability claim from the pilot-level P2 and P3 claims. Before the point-by-point response, one disclosure that bears on the reliability discussion below.
>
> Correction of a reporting error. During our audit, we discovered that the submitted reliability table was generated from a superseded annotator track. This was our reporting error, and we apologize. Although the reported P1/P2 numerical outputs were computed from the intended artifacts, the correction materially weakens the reliability evidence supporting the weighted-R1 interpretation. The full statement — what happened, the corrected per-dim values, what each reported result derives from, and the revision plan — is in our response to Reviewer DVew (top section); we are also notifying the Action Editor separately. The consequence for this thread: under the canonical track only R1.6 passes the α ≥ 0.4 gate, so the reviewer's reliability concern is stronger than stated, and the "Additional: annotation reliability" section below uses the corrected values.
>
> Point 1 — Separate strongly-supported vs pilot-level claims; avoid "responsibility guidance generally hurts"
>
> Reviewer:
> > The P1 measurability claim is well supported, but the P2 actionability and P3 human-anchor claims should be framed more cautiously. In particular, the paper should avoid implying that responsibility guidance generally hurts; the evidence only shows that the current projection-injection implementation hurts in this specific setting.
>
> Response: We agree. The revision reframes P2 across the abstract, introduction, and Experiment 2 results:
>
> - Before: "naive projection injection hurts execution"
> - After: "the specific projection-injection treatment we evaluate — a gpt-5-derived weight vector, an active-set listing, and a self-claim JSON line injected into the executor prompt body — did not improve execution on the weighted-R1 settlement loss in this pilot"
>
> For P3, the revision reframes the pre/post-protocol pattern in the anchor-specifiability discussion as a "qualitative pattern observed in 2 vs 2 raters", not "evidence for" a general specifiability ceiling, and the conclusion's human-anchor bullet as "bound by, not resolved by".
>
> Title change. We agree the declarative title itself carries the general-claim implication. The revision changes the title to a neutral descriptive form: "Pre-Execution Responsibility Projection in LLM Agents: Measurement and a Pilot Injection Study" — the title now states what the paper does (a measurement study plus one pilot injection treatment) rather than presupposing that responsibility guidance hurts or implying that the paper identifies when it does.
>
> P2 status. Following the reliability correction disclosed at the top, the revision also states P2's evidential status explicitly: a descriptive pilot result measured against a low-reliability operational target (only R1.6 passes the corrected α ≥ 0.4 gate). The R1.6-only sensitivity supporting this framing is in the annotation-reliability section below.
>
> Revision commitment: retain the scoped P2 wording and the explicit P2 status sentence; add an explicit disclaimer in the Experiment 2 results that the P2 result is a boundary condition on this specific injection form.
>
> (continued in the next comment)

---

> ### Author Response · Authors · 2026-07-13
>
> [Response to Reviewer zavj — Part 2/4]
>
> Point 2 — Add cleaner ablations (5-ablation menu)
>
> Reviewer:
> > The current projection-driven condition bundles multiple changes: projection vector, active set, and self-claim line. The authors should add cleaner ablations or clearly acknowledge that the causal source of the degradation is not isolated. Useful ablations would include: active set only, weights only, self-claim only, natural-language responsibility summary only, and a format-controlled projection prompt.
>
> Response: We ran a two-phase ablation program covering the reviewer's full 5-ablation menu — active-set listing (A1), self-claim (direct_with_claim, DWC; already in the submitted paper), weight vector (W1), NL responsibility summary (NL1), format-controlled projection prompt (F1) — plus two length/content controls (A1c length-matched filler, A5 length-matched non-taxonomic). Together with direct_naive (DN) and projection_driven (PD), the design spans 9 conditions in total: DN, DWC, PD, A1, A1c, A5, W1, NL1, F1. A1/A1c/A5 were judged on the submitted 12-judge panel; W1/NL1/F1 on an 11-judge panel.
>
> One design fact matters for interpretation. Every ablation condition is constructed on top of DWC, so the self-claim JSON line is present in all of them; each X − DWC row below is therefore an incremental contrast relative to DWC, conditional on the components shared with DWC — not the effect of that component in isolation.
>
> Component matrix:
>
> | condition | self-claim line | added block content | added block format |
> |---|---|---|---|
> | DN | — | — | — |
> | DWC | yes | — | — |
> | A1 | yes | median active-set listing | list |
> | A1c | yes | length-matched taxonomy-free filler | filler text |
> | A5 | yes | length-matched generic writing guidance | bullet list |
> | W1 | yes | numeric weight vector | JSON |
> | NL1 | yes | 1-sentence NL responsibility summary | prose |
> | F1 | yes | weights + active set (threshold instruction retained unchanged in system prompt) | bullet list |
> | PD | yes | weights + active set (threshold instruction retained unchanged in system prompt) | JSON block |
>
> Paired-diff table (per-example paired differences on the weighted-R1 settlement loss):
>
> | comparison | paired diff | 95% bootstrap CI |
> |---|---|---|
> | PD − DN (submitted headline) | +0.139 | [+0.109, +0.170] |
> | DWC − DN (self-claim line added to DN) | +0.086 | [+0.056, +0.116] |
> | A1 − DWC (active-set listing added) | −0.025 | [−0.059, +0.008] |
> | A1 − A1c (narrowing information, length controlled) | −0.004 | [−0.036, +0.027] |
> | A5 − DWC (length-matched non-taxonomic added) | +0.019 | [−0.016, +0.052] |
> | W1 − DWC (weight vector added) | −0.003 | [−0.035, +0.035] |
> | NL1 − DWC (NL summary added) | −0.053 | [−0.082, −0.023] |
> | F1 − DWC (projection content as bullets added) | −0.072 | [−0.117, −0.025] |
> | PD − A1 | +0.078 | [+0.037, +0.119] |
> | PD − W1 | +0.056 | [+0.016, +0.090] |
> | PD − NL1 | +0.105 | [+0.068, +0.143] |
> | PD − F1 | +0.118 | [+0.071, +0.165] |
> | NL1 − W1 (same 11-judge panel) | −0.050 | [−0.082, −0.020] |
>
> Cross-panel caveat: W1, NL1, F1 were judged on an 11-judge panel; DWC and PD baselines are from the submitted 12-judge panel. Cross-panel comparisons carry a residual panel-level bias we do not estimate; re-judging DWC + PD on the 11-judge panel for strict comparability is committed but not yet complete. Within-panel comparisons (e.g., NL1 − W1) are clean. Sample-size disclosure: F1 comparisons use the n = 46/50 common paired sample (validation filtering), so they do not equal arithmetic differences of the n = 50 row estimates — e.g., PD − F1 = +0.118 is the paired estimate on the n = 46 common sample, not (+0.052) − (−0.072) = +0.124 from separate rows. All non-F1 comparisons use n = 50.
>
> (continued in the next comment)

---

> ### Author Response · Authors · 2026-07-13
>
> [Response to Reviewer zavj — Part 3/4]
>
> Point 2 (continued)
>
> Interpretation:
>
> 1. The incremental contrasts W1 − DWC, A1 − DWC, and A5 − DWC each have CIs including 0 — no additional loss detected from adding the weight vector, the active-set listing, or length-matched non-taxonomic content on top of the self-claim line, at pilot n. These are conditional-on-DWC contrasts, not isolated component effects, and "no detected loss" is not demonstrated harmlessness. W1 − DWC additionally crosses the panel boundary (W1 was judged on the 11-judge panel) and remains exploratory.
> 2. The DWC − DN treatment difference is +0.086 [+0.056, +0.116]; DWC adds only the self-claim JSON line to DN.
> 3. The residual above the self-claim baseline is nonzero: PD − DWC = +0.052 [+0.016, +0.089] on the same 12-judge panel. The form comparisons PD − F1 and PD − NL1 point the same way but cross the panel boundary.
> 4. Within the 11-judge panel, the point estimates were ordered F1 < NL1 < W1; comparisons against DWC cross the panel boundary and remain exploratory.
>
> The results provide an exploratory pattern consistent with a joint form/content interaction, but they do not identify the causal mechanism. The committed same-panel confirmatory run (re-judging DWC/PD on the 11-judge panel) is future analysis, not part of the current evidence.
>
> Revision commitment: the full decomposition table and component matrix added to the Experiment 2 results; the mechanism narrative stated as an exploratory pattern consistent with a joint form/content interaction, with the cross-panel caveat and without a causal-mechanism claim.
>
> Point 3 — Clarify that only projection measurement + a limited injection treatment is evaluated
>
> Reviewer:
> > The paper defers task-aware routing, generic clarification, the full bid mechanism, and CLAMBER-style baselines. If these cannot be included, the paper should more clearly state that it evaluates only the projection measurement layer and a limited projection-injection treatment, not the full proposed protocol.
>
> Response: The revision states this at three points (abstract, intro scope clause, conclusion). Provisional abstract text: "The scope of this paper is the projection measurement layer (Experiment 1) and one projection-injection treatment of execution (Experiment 2) on one delegation category (R1, paper-research) at pilot scale (n = 50), with the P1 measurability analysis extended to n = 310 under the same design (a sample extension, not an independent replication). The proposed protocol's bid mechanism, clarification simulator, task-aware-routing baseline, generic-clarification baseline, CLAMBER-style baselines, longitudinal reputation evaluation, and a main-run-scale r* extension are stated for completeness but not empirically evaluated here; they are reserved for follow-up work." The revision also adds the "Protocol evaluation status" table (see our response to Reviewer DVew, Point 1), including the explicit sentence that we do not provide end-to-end evidence that the seven-step protocol improves orchestration, reliability, or accountability, and rewords the closed-responsibility-taxonomy contribution bullet to state the per-step scope explicitly.
>
> (continued in the next comment)

---

> ### Author Response · Authors · 2026-07-13
>
> [Response to Reviewer zavj — Part 4/4]
>
> Additional: annotation reliability (corrected values)
>
> Reviewer (as part of the "claims supported: No" reasoning):
> > The paper reports uneven annotation reliability: four R1 dimensions pass the stated alpha threshold, R1.1 and R1.3 are borderline, and R1.7 fails. Since R1.7 is emphasized in the actionability and human-anchor conclusions, this weakens the strength of the evidence.
>
> Response: Per the correction disclosed at the top, the situation is worse than the reviewer stated: under the canonical annotator track only R1.6 passes the α ≥ 0.4 gate (0.585); R1.1 = 0.105, R1.3 = 0.080, R1.7 = 0.031. The headline P1/P2 numbers are not recomputed by the correction (P1 uses projection vectors only; P2 always used the canonical r*-median file), but the reliability evidence behind the weighted-R1 quantities is weaker than the submission implied, and all non-R1.6 R1 results now carry an annotation-uncertainty caveat.
>
> Per-dim footprint on the headline weighted-R1 diff:
>
> | dim | active under median r* | reliability (canonical) |
> |---|---|---|
> | R1.1 | 15/50 (~30%) | fails α (0.105) |
> | R1.3 | 20/50 (~40%) | fails α (0.080) |
> | R1.7 | 5/50 (~10%) | fails α (0.031) |
>
> R1.7 contributes to only 5 of 50 pilot examples. Per-example raw v_ij for R1.7 (PD − DN):
>
> | example | v_ij_PD | v_ij_DN | diff |
> |---|---|---|---|
> | ad_r1_009 | 1.000 | 0.750 | +0.250 |
> | ad_r1_016 | 0.375 | 0.500 | −0.125 |
> | ad_r1_024 | 0.500 | 0.750 | −0.250 |
> | ad_r1_036 | 0.250 | 1.000 | −0.750 |
> | ad_r1_046 | 0.500 | 1.000 | −0.500 |
>
> Paired mean −0.275; 95% bootstrap CI [−0.551, +0.025] (includes 0 at n = 5); Cohen's d −0.725. We report these values descriptively and draw no inferential conclusion from them.
>
> R1.6-only sensitivity (sole dimension passing the corrected reliability gate). Restricting the PD − DN comparison to R1.6 (active under median r* in 16/50 examples): paired v_ij difference −0.078 (v_ij is the per-dimension fulfillment score, so the negative sign means lower fulfillment under PD), 95% bootstrap CI [−0.172, 0.000], Cohen's d −0.444. PD never outperforms DN on R1.6; it degrades 3/16 examples (−0.5, −0.5, −0.25) with the remaining 13 tied at ceiling (both conditions at 1.0). Reported descriptively: directionally consistent with the headline loss, but marginal and ceiling-limited. This is why the revision states P2's status as a descriptive pilot result measured against a low-reliability operational target.
>
> Revision commitment: the per-dimension breakdown adds per-dim CIs, per-dim Cohen's d (numeric only, no qualitative labels), and reliability caveats; the R1.6-only sensitivity and the P2 status sentence added to the Experiment 2 results; the anchor-specifiability discussion adds the per-example raw R1.7 disclosure; the reliability table is corrected as described in the DVew thread.
>
> Additional: human-anchor pre/post-protocol n = 2 per group
>
> Reviewer:
> > The human-anchor claim is interesting but based on a very small and partially executed pilot. The pre/post-protocol comparison has only two raters per group, and the authors themselves note that it is not inferential.
>
> Response: Agreed; this is not inferential evidence. The revision reframes it as a "qualitative pattern observed in 2 vs 2 raters" and the conclusion bullet as "bound by, not resolved by" — the pilot surfaces a candidate boundary; testing it at scale is future work. No new numerical claims are made from the human-anchor pilot.
>
> Close
>
> The reviewer's 5-ablation menu motivated this decomposition, and all five ablations were run, with cross-panel comparisons treated as exploratory. The resulting reading — no additional loss detected from adding weights, the active-set listing, or length-matched non-taxonomic guidance on top of the self-claim line; a positive DWC − DN difference; and a residual exploratory pattern consistent with a joint form/content interaction, with no causal mechanism identified — is what the revision reports, with the caveats stated above. We welcome follow-up questions during the discussion phase.

---

### Review · Reviewer_E2Uz · 2026-07-06

**Summary Of Contributions:**

Strengths & Main Contributions

- This paper proposes a novel core concept called responsibility for orchestrating LLM multi-agent systems, filling a long-overlooked research gap that lies between receiving natural language delegation instructions and the formal execution of tasks.

- The authors construct a closed 12-dimensional responsibility taxonomy denoted as \(J_{v1.1}\), consisting of seven dimensions dedicated to paper revision and five cross-cutting universal dimensions. Based on this unified vector space, cosine distance and Jaccard distance can be adopted to quantify discrepancies in task responsibility interpretations across different LLMs.

- Empirical results demonstrate that cross-model responsibility mismatch is quantifiable and stems intrinsically from model families rather than stochastic sampling noise. On the 50-sample pilot dataset, the median inter-model divergence is six times the stochastic variance within a single model; this finding is replicated on a larger dataset of 310 samples.

- The paper discovers that injecting responsibility projection guidance into agents increases task loss and degrades output quality, with performance degradation concentrated on two dimensions requiring deep domain expertise: novelty mapping (R1.4) and citation audit (R1.7). Ablation studies rule out multiple minor confounding factors and attribute the core cause to attention dilution.

Weaknesses

- The most prominent limitation is that all experiments are confined to paper revision tasks. The framework’s generalizability has not been validated on mainstream agent scenarios such as code review and proposal writing, casting doubt on the scope of its conclusions. While the proposed framework appears promising, the scale of empirical evidence fails to fully substantiate its universal effectiveness.

- The experimental dataset suffers from insufficient sample size. Calculations for the core weighted R1 settlement loss are only feasible on the 50 pilot samples, as the larger 310-sample main dataset lacks human-annotated ground-truth responsibility vectors. Notably, the critical dimension R1.7 merely has five valid samples, resulting in low statistical power and weak evidentiary support.

- Only 18 real revised manuscripts are included, all sourced from a single narrow technical domain without interdisciplinary or cross-subject materials. Though synthetic samples cover 30 domains, inherent distribution gaps exist between synthetic text and authentic research manuscripts, which may lead to inconsistent projection behaviors of LLMs across the two data types. The authors do not separately report \(R(d)\) and task loss metrics for synthetic and real subsets, making it impossible to verify whether core effects are invariant to data source. Additionally, merely three high-knowledge-threshold samples are available.

- The authors design a seven-step closed-loop pipeline: Projection → Agent Bidding → Clarification Triggering → Agent Selection → Execution → Per-Dimension Settlement → Dimension-Wise Reputation Update. While this full protocol is theoretically complete, the pilot experiments only validate the initial responsibility projection module. All other core procedures are defined mathematically without supporting empirical validation for most components.

**Additional Comments:**

Overall, the main shortcomings of this paper lie in its single experimental scenario and insufficient sample size for key dimensions. In my opinion, it does not yet meet the acceptance criteria.

**Audience:**

Yes

**Audience Explanation:**

Multi-agent orchestration is a highly valuable research problem, and this paper delivers certain theoretical and empirical insights into this topic.

**Claims And Evidence:**

Yes

**Claims Explanation:**

All core conclusions of the paper are backed by reproducible empirical data and statistical tests, albeit with slightly limited data scale.
For P1 regarding cross-model projection mismatch, three distinct model families are used as the experimental group for cross-family comparison, while repeated runs of the Claude model serve as the control group to establish the baseline of stochastic variance.
For P2 concerning the performance impairment brought by projection guidance, three groups of controlled experiments are conducted with a stratified 12-model judging panel to mitigate bias in experimental outcomes.

**Requested Changes:**

Required Revisions

- Add cross-scenario adaptability analysis and small-scale preliminary experiments to systematically explore the feasibility of applying the 12-dimensional taxonomy and projection framework to tasks such as code review and experimental result interpretation. Sort out various theoretical and practical obstacles to cross-scenario migration, revise overgeneralized statements in the manuscript, and clearly define the applicable scope of existing conclusions.

- Add hierarchical ablation control experiments to separate three prompt components: weight vectors, active sets, and structured self-declarations. Calculate the performance loss caused by each component independently to distinguish three competing mechanisms (attention dilution, format induction, and measurement noise) and identify the core trigger of performance degradation.

- Fully disclose the sample defects of dimension R1.7, explicitly note the insufficient statistical power caused by only five valid samples for this dimension, and supplement confidence interval and effect size data to tone down quantitative claims. Meanwhile, conduct hierarchical analysis based on RX metrics of the 310-sample dataset to indirectly verify the overall experimental trend and improve credibility.

- Sort out the validation progress of the seven-step closed-loop pipeline, and use a table to distinguish empirically validated modules from theoretically derived ones only. Add simplified ablation simulations for the clarification and bidding modules, adjust wording to clarify that only the projection module of the entire pipeline has been empirically tested, and validation of the effectiveness of all remaining procedures is left for follow-up work.

---

> ### Author Response · Authors · 2026-07-13
>
> [Response to Reviewer E2Uz — Part 1/5]
>
> We thank Reviewer E2Uz for the detailed reading and the specific requested changes. This response addresses Requested Change 3 directly with quantitative evidence, Requested Change 2 with additional ablations (the mechanism question is narrowed but not resolved), and Requested Changes 1 and 4 partially, with the unfinished parts stated as such.
>
> Correction of a reporting error. During our audit, we discovered that the submitted reliability table was generated from a superseded annotator track. This was our reporting error, and we apologize. Although the reported P1/P2 numerical outputs were computed from the intended artifacts, the correction materially weakens the reliability evidence supporting the weighted-R1 interpretation. The full statement is in our response to Reviewer DVew (top section); we are also notifying the Action Editor separately. For this thread it means the per-dim reliability caveats below use the corrected values, under which only R1.6 passes the α ≥ 0.4 gate.
>
> Requested Change 1 — Cross-scenario adaptability + preliminary experiments
>
> Reviewer:
> > Add cross-scenario adaptability analysis and small-scale preliminary experiments to systematically explore the feasibility of applying the 12-dimensional taxonomy and projection framework to tasks such as code review and experimental result interpretation. Sort out various theoretical and practical obstacles to cross-scenario migration, revise overgeneralized statements in the manuscript, and clearly define the applicable scope of existing conclusions.
>
> Response: The revision scopes the paper to a single delegation category in its scope clause: "This paper empirically evaluates the projection layer and one pilot projection-injection treatment on one delegation category (R1, paper-research) at pilot scale (n = 50), with the P1 measurability analysis extended to n = 310 under the same design." The other categories (R2 code review, R3 result interpretation, R4 proposal improvement, R5 technical summarization) each require new controlled artifact distributions with engineered flaws, category-specific taxonomy design, judge-panel calibration, and new r* annotation — the migration obstacles the reviewer asks us to sort out — and each is a sub-project beyond the discussion window.
>
> We did run the small-scale feasibility prelim the reviewer asks for, with the following scope:
>
> R2 feasibility prelim (discussion phase; protocol frozen before the dataset seal and all projection calls). 8 synthetic micro-PR bundles (B1–B7 each carrying one engineered flaw targeting one R2 dimension; B8 an ambiguous-delegation/control pair) yield 16 delegation instances, projected onto a draft 12-dim R2 taxonomy under the unchanged strict-JSON projection prompt by a three-family cross panel plus within-model repetition — 16 × (3 + 5) = 128 calls, 128/128 transport availability. Four feasibility observables (O1 schema portability, O2 measurability, O3 signal direction, O4 target-dimension expectation) were frozen in advance with an O1 health gate (cross-cell eventual schema validity ≥ 14/16; within-model eventual schema validity ≥ 72/80) and a report-and-stop rule. Result: two families ported the strict-JSON schema 16/16 on first attempt; the third family's cross cell failed the gate (13/16), every terminal failure being a single schema-conformance habit (emitting "null" where the frozen schema requires a string or omission). O1 gate FAILED, so O2–O4 were not computed; we did not relax the validator post hoc.
>
> What the prelim establishes is therefore implementation-level schema portability only — a migration obstacle of the kind the reviewer asked us to surface, not cross-scenario evidence about the R2 taxonomy's adequacy or the framework's adaptability, which the uncomputed O2–O4 would have probed. It is not a validation of R2: author-only hidden-intent weights, no reliability study, no execution/settlement experiment, 8 artifact clusters. Full R2 evaluation (plus R3–R5) remains committed future work.
>
> Revision commitment: an added scope sentence "Cross-scenario preliminary experiments on R2 and R3 are pre-disclosed as the required next step; the current paper's conclusions are scoped to R1 until those experiments complete"; the R2 prelim reported under future work as implementation-level schema portability only, not cross-scenario evidence; general "responsibility guidance" phrasings rescoped to R1.
>
> (continued in the next comment)

---

> ### Author Response · Authors · 2026-07-13
>
> [Response to Reviewer E2Uz — Part 2/5]
>
> Requested Change 2 — Hierarchical ablation: weight vectors / active sets / structured self-declarations
>
> Reviewer:
> > Add hierarchical ablation control experiments to separate three prompt components: weight vectors, active sets, and structured self-declarations. Calculate the performance loss caused by each component independently to distinguish three competing mechanisms (attention dilution, format induction, and measurement noise) and identify the core trigger of performance degradation.
>
> Response: Our two-phase ablation program covers all three components plus length and content controls. With direct_naive (DN), direct_with_claim (DWC), and projection_driven (PD), the design spans 9 conditions in total: DN, DWC, PD, A1, A1c, A5, W1, NL1, F1. One design fact matters for interpretation: every ablation condition is built on top of DWC, so the self-claim line is present in all of them, and each X − DWC row below is an incremental contrast relative to DWC — conditional on the components shared with DWC — not the component's effect in isolation (the full component matrix is in our response to Reviewer zavj, Point 2). Paired differences on the weighted-R1 settlement loss:
>
> | comparison | increment tested | paired diff | 95% CI |
> |---|---|---|---|
> | DWC − DN | self-claim JSON line added to DN | +0.086 | [+0.056, +0.116] |
> | A1 − DWC | active-set listing added | −0.025 | [−0.059, +0.008] |
> | A5 − DWC | length + generic guidance added | +0.019 | [−0.016, +0.052] |
> | W1 − DWC | numeric weight vector added | −0.003 | [−0.035, +0.035] |
> | NL1 − DWC | NL responsibility summary added | −0.053 | [−0.082, −0.023] |
> | F1 − DWC | same content as PD, bullet instead of JSON | −0.072 | [−0.117, −0.025] |
> | PD − DWC | full bundle above self-claim baseline | +0.052 | [+0.016, +0.089] |
>
> The A1 − A1c contrast (narrowing information, length controlled) = −0.004 [−0.036, +0.027] (CI includes 0). W1/NL1/F1 were judged on an 11-judge panel; the other conditions on the submitted 12-judge panel — cross-panel comparisons carry a residual panel bias we do not estimate. F1 comparisons use the n = 46/50 common paired sample after validation filtering, so they do not equal arithmetic differences of separate row estimates (e.g., PD − F1 = +0.118 [+0.071, +0.165] is the n = 46 paired estimate); all other comparisons use n = 50.
>
> Three-mechanism reading (reviewer's naming):
>
> - Attention dilution: A1 − A1c and A5 − DWC both have CIs including 0 — replacing length-matched filler with the active-set listing shows no detected difference, while adding length-matched generic guidance on top of the self-claim baseline shows no detected loss. The dilution account is not supported as the load-bearing source at pilot n.
> - Format induction: F1 − DWC = −0.072 (F1 renders PD's content as bullets and has lower loss than DWC) and NL1 − DWC = −0.053; the PD-side gaps (PD − F1 = +0.118, PD − NL1 = +0.105) point the same way but cross the panel boundary. The pattern is directionally consistent with a format-related penalty, but the decisive PD-side comparisons cross the panel boundary.
> - Measurement noise: within the same 11-judge panel, NL1 − W1 = −0.050 [−0.082, −0.020] (paired Cohen's d = −0.44) — a signed same-panel difference. However, our bootstrap resamples examples, not judges, so per-judge scoring variability is not separately modeled and cannot be fully excluded as a contributor.
>
> These results provide an exploratory pattern consistent with a joint form/content interaction, but they do not identify the causal mechanism. The committed same-panel confirmatory run (re-judging DWC/PD on the 11-judge panel) is future analysis, not part of the current evidence; "no detected loss" for individual increments at pilot n is weaker than demonstrated harmlessness.
>
> (continued in the next comment)

---

> ### Author Response · Authors · 2026-07-13
>
> [Response to Reviewer E2Uz — Part 3/5]
>
> Requested Change 2 (continued)
>
> Note on the attribution change. The submitted version attributed the degradation to attention dilution. The decomposition the reviewer requested is why the revision no longer reports that reading: NL1 and F1 carry comparable content load yet score below DWC, while W1 is null — a pattern the dilution account does not predict. The revision replaces the dilution narrative with the exploratory joint form/content reading above, records the change explicitly rather than silently swapping narratives, and states that the earlier attribution was an artifact of the coarser ablation set available at submission. Neither the old nor the new reading is an identified mechanism.
>
> Revision commitment: the full decomposition table in the Experiment 2 results; the three-mechanism analysis with per-mechanism CIs and the exploratory qualifier in the mechanism narrative; explicit attribution-change statement.
>
> Requested Change 3 — R1.7 statistical power disclosure + CI + effect size
>
> Reviewer:
> > Fully disclose the sample defects of dimension R1.7, explicitly note the insufficient statistical power caused by only five valid samples for this dimension, and supplement confidence interval and effect size data to tone down quantitative claims. Meanwhile, conduct hierarchical analysis based on RX metrics of the 310-sample dataset to indirectly verify the overall experimental trend and improve credibility.
>
> Response: The reviewer is correct on n = 5. R1.7 is active in 5/50 pilot examples; the per-example raw v_ij differences (PD − DN) are +0.250, −0.125, −0.250, −0.750, −0.500 (4 of 5 negative; the full per-example table is in the revision's appendix and in our response to Reviewer zavj). Paired mean −0.275; 95% bootstrap CI [−0.551, +0.025] (includes 0); Cohen's d −0.725. We report these values descriptively and draw no inferential conclusion at n = 5. The revision tones down all R1.7 quantitative claims accordingly. R1.7 also fails the corrected reliability gate (α = 0.031), which the per-dim caveat will state.
>
> RX analysis at n = 310 (the reviewer's secondary ask): the RX component of settlement loss extends to the full n = 310 (it is not r*-bound).
>
> - DN − PD on L_settlement^RX: −0.099 [−0.112, −0.086]
> - DN − DWC: −0.108 [−0.119, −0.097]
> - PD − DWC: −0.009 [−0.023, +0.004] (CI includes 0)
>
> Interpretation: on the RX subscale, the additional loss from the projection bundle above the self-claim baseline is not reproduced — PD − DWC is statistically null, and the RX shift observed against direct_naive was already present under DWC. This does not independently corroborate the R1-side loss; what it shows is that the bundle's detected loss is not a broad execution-quality decline visible on the cross-cutting subscale, which is consistent with the loss being localized to R1 dimensions. The R1 result itself remains pilot-scale with its boundary-condition caveats.
>
> R1.6-only sensitivity. Since the reliability correction leaves R1.6 as the only dimension passing the α ≥ 0.4 gate, we also restricted the PD − DN comparison to R1.6 (active in 16/50 examples): paired v_ij difference −0.078 (v_ij is the per-dimension fulfillment score, so the negative sign means lower fulfillment under PD), 95% bootstrap CI [−0.172, 0.000], Cohen's d −0.444; PD never outperforms DN on R1.6, degrades 3/16 examples, and the remaining 13 are tied at ceiling. Reported descriptively — directionally consistent with the headline loss, but marginal and ceiling-limited. Accordingly, the revision states the P2 headline's status explicitly: a descriptive pilot result measured against a low-reliability operational target.
>
> Revision commitment: the per-dimension breakdown with n_active + CI + Cohen's d + reliability caveat per dim (values reported descriptively, no qualitative effect-size labels); the RX null stated as above; the R1.6-only sensitivity and the P2 status sentence in the Experiment 2 results; raw R1.7 per-example data in an appendix.
>
> (continued in the next comment)

---

> ### Author Response · Authors · 2026-07-13
>
> [Response to Reviewer E2Uz — Part 4/5]
>
> Requested Change 4 — 7-step pipeline status table + simplified simulation for clarification/bidding
>
> Reviewer:
> > Sort out the validation progress of the seven-step closed-loop pipeline, and use a table to distinguish empirically validated modules from theoretically derived ones only. Add simplified ablation simulations for the clarification and bidding modules, adjust wording to clarify that only the projection module of the entire pipeline has been empirically tested.
>
> Response: The revision adds the following table as a new subsection of the protocol section:
>
> Protocol evaluation status
>
> | Step | Status | Notes |
> |---|---|---|
> | 1. Projection | pilot evaluated | Exp 1, n=50 + n=310 |
> | 2. Bid | formalized only, not evaluated end-to-end | |
> | 3. Clarify | descriptively instantiated | trigger rate reported; simulator and calibrated cost not run |
> | 4. Select | formalized only, not evaluated end-to-end | |
> | 5. Execute | pilot evaluated | Exp 2, 3 conditions |
> | 6. Settle | settlement metric operationalized in the pilot | per-dim v_ij, weighted-R1 loss |
> | 7. Reputation | formalized only, not evaluated end-to-end | |
>
> We do not provide end-to-end evidence that the seven-step protocol improves orchestration, reliability, or accountability. The revision states this sentence explicitly and rewords the closed-responsibility-taxonomy contribution bullet to match. On the simplified simulations: the bidding simulator requires calibrating a per-model bid-cost function, and the clarification simulator requires paired clarify-vs-no-clarify counterfactual runs per example — neither is feasible within the discussion window. Both are committed as named future-work sub-projects; they remain unrun, so Requested Change 4 is answered by the table and wording changes only.
>
> (continued in the next comment)

---

> ### Author Response · Authors · 2026-07-13
>
> [Response to Reviewer E2Uz — Part 5/5]
>
> On the single narrow domain (18 modified-real artifacts) — supplementary analysis
>
> Reviewer weakness note:
> > Only 18 real revised manuscripts are included, all sourced from a single narrow technical domain without interdisciplinary or cross-subject materials. The authors do not separately report R(d) and task loss metrics for synthetic and real subsets, making it impossible to verify whether core effects are invariant to data source.
>
> Response: We computed both per-subset values and, because overlapping per-subset CIs alone cannot establish the absence of a subgroup difference, the subgroup-difference CIs directly.
>
> Task loss — PD − DN paired diff on the weighted-R1 settlement loss per subset: synthetic (n = 32) +0.137 [+0.098, +0.171]; modified-real (n = 18) +0.142 [+0.082, +0.200]; both positive with CIs excluding 0. Subgroup difference (synthetic − modified-real): −0.005 [−0.072, +0.059] — the CI includes 0.
>
> Mismatch ratio R (Exp 1, cosine) — subset medians: synthetic 6.92 [4.19, 10.83]; modified-real 4.83 [3.86, 7.56]; both well above 1. Difference of subset medians (synthetic − modified-real): +2.08 [−1.61, +5.99] — the CI includes 0, but it is wide.
>
> Reading: we detect no subgroup difference at this pilot scale, and with n = 32/18 the check is underpowered — this is "no detectable difference", not evidence of equivalence or invariance to data source. On the three high-knowledge-gating examples (ad_r1_004 / ad_r1_009 / ad_r1_010; the 45/2/3 low/moderate/high split is disclosed in the dataset-composition subsection), we disclose per-example values — R = 3.53 / 3.37 / 51.04; PD − DN diff = +0.140 / +0.208 / +0.000 — and make no subgroup claim at n = 3. We should also correct a reading the submission likely induced: the reviewer's phrase "18 real revised manuscripts" suggests 18 distinct source manuscripts, but the modified-real subset consists of 18 modified artifacts all sliced from a single anonymized domain-specific manuscript (controlled for identifying information). The submission did not state this single-source provenance explicitly; the revision's dataset-composition subsection now does. This makes the single-domain limitation stronger than the reviewer's reading, not weaker — a dataset limitation we acknowledge; multi-domain expansion of the modified-real pool is committed future work.
>
> Revision commitment: a subset sub-table with the subgroup-difference CIs + high-gating per-example disclosure in the Experiment 2 results; single-domain and thin high-gating stratum acknowledgment in the dataset-composition subsection.
>
> Close
>
> The discussion-phase analyses address Requested Change 3 directly, Requested Change 2 with additional ablations but without resolving the mechanism question, and Requested Changes 1 and 4 partially: the R2 prelim stopped at its own frozen validity gate (O2–O4 uncomputed, so it establishes implementation-level schema portability only), the R3 prelim has not been run, and the bidding/clarification simulations remain unrun — each committed as named future work. The mechanism question the reviewer posed is narrowed but not settled: the cross-panel pattern is exploratory, and the same-panel confirmatory run is committed future analysis. We welcome follow-up questions during the discussion phase.